# An embryonic stem cell-specific heterochromatin state promotes core histone exchange in the absence of DNA accessibility

Carmen Navarro [1,2], Jing Lyu [1,2,3], Anna-Maria Katsori [1,2,3], Rozina Caridha[1,2] & Simon J. Elsässer [1,2✉]

Nucleosome turnover concomitant with incorporation of the replication-independent histone variant H3.3 is a hallmark of regulatory regions in the animal genome. Nucleosome turnover is known to be universally linked to DNA accessibility and histone acetylation. In mouse embryonic stem cells, H3.3 is also highly enriched at interstitial heterochromatin, most prominently at intracisternal A-particle endogenous retroviral elements. Interstitial heterochromatin is established over confined domains by the TRIM28-KAP1/SETDB1 corepressor complex and has stereotypical features of repressive chromatin, such as H3K9me3 and recruitment of all HP1 isoforms. Here, we demonstrate that fast histone turnover and H3.3 incorporation is compatible with these hallmarks of heterochromatin. Further, we find that Smarcad1 chromatin remodeler evicts nucleosomes generating accessible DNA. Free DNA is repackaged via DAXX-mediated nucleosome assembly with histone variant H3.3 in this dynamic heterochromatin state. Loss of H3.3 in mouse embryonic stem cells elicits a highly specific opening of interstitial heterochromatin with minimal effects on other silent or active regions of the genome.

---

[1] Science for Life Laboratory, Department of Medical Biochemistry and Biophysics, Karolinska Institutet, Tomtebodavägen 23, 17165 Stockholm, Sweden. [2] Ming Wai Lau Centre for Reparative Medicine, Stockholm node, Karolinska Institutet, Solnavägen 9, 17165 Stockholm, Sweden. [3]These authors contributed equally: Jing Lyu, Anna-Maria Katsori. ✉email: simon.elsasser@scilifelab.se

Histone H3.3 is an evolutionary conserved variant of the canonical histone H3 proteins, termed H3.1 and H3.2 in mammalian cells[1]. First described to associate with regions of active transcription in *Drosophila*[2], H3.3 has been recognized as the sole substrate for replication-independent chromatin assembly in mammalian cells[3]. While H3.1/2 expression and replication-dependent incorporation is limited to S phase, H3.3 is expressed throughout the cell cycle and incorporated at sites of dynamic nucleosome turnover. A multitude of genome-wide studies have identified such dynamic chromatin at promoters, enhancers, gene bodies and origins of replication[4–13]. In these instances, nucleosome turnover is thought to be a consequence of chromatin remodeling, transcriptional activity, nucleosome-destabilizing DNA sequences, competition with other chromatin binding factors, or a combination thereof. While histone H3.3 itself has been proposed to destabilize the nucleosome, it is unclear if H3.3 has a causal role in promoting nucleosome turnover[14–16]. Histone H3.3 has been shown to be enriched in histone posttranslational modifications (PTMs) characteristic of euchromatin, i.e. H3K4me3, H3K9ac, H3K27ac[17,18]. Accumulation of active PTMs on histone H3.3 is sufficiently explained by their co-occurrence at sites of active enhancers or transcription, and H3.3 appears not to be required for maintaining those active PTMs. However, a notable exception represents phosphorylation of Ser31, a H3.3-specific residue. H3.3Ser31 phosphorylation facilitates rapid activation of genes[19,20], thus providing a mechanistic link between H3.3 incorporation and gene activation. In summary, histone H3.3 is a well-established component of euchromatin, acting as a replacement histone for a range of dynamic processes. While H3.3 has to be considered predominantly neutral to the underlying dynamic process, it can – as exemplified in the specific instance above – partake mechanistically in establishing open chromatin.

In mouse embryonic stem cells (ESC), a considerable fraction of H3.3-enriched regions does not fall into the active regions outlined above, but colocalizes with histone H3 Lys 9 trimethylation (H3K9me3) modification at interstitial heterochromatin[21]. Re-ChIP experiments suggest that H3.3 and H3K9me3 coincide in the same nucleosome[21]. Interstitial heterochromatin in mouse ESC spreads over relatively small genomic distance (~10 kb), is established by a Tripartite motif-containing 28 (TRIM28 also known as KAP1, TIF1b) co-repressor/SETDB1 histone methyltransferase complex and includes a subset of endogenous retroviral elements (ERVs), most prominently intracisternal A-type particle (IAP), and imprinted genes[22–27]. KAP1 is recruited to foreign DNA elements through numerous DNA sequence specific Krüppel-associated box containing Zinc-Finger proteins (KRAB-ZFPs).

Interstitial heterochromatin is highly CpG-methylated, enriched in linker histone and bound by HP1 family proteins, the stereotypical H3K9me3 readers[28–30]. HP1 proteins have been shown to bridge nucleosomes and compact chromatin into phase-separated liquid condensates[31,32]. HP1 and linker histone render DNA in an inaccessible state[32], and underlying genes/repeats are refractory to activating factors[33].

Enrichment of H3.3 at H3K9me3 heterochromatin regions, so far uniquely observed in mouse ESC, poses a conundrum: do these regions show fast nucleosome turnover as known for euchromatic H3.3-enriched regions? Does the presence of H3.3 coincide with DNA accessibility, and how would such dynamic properties be reconciled with a silent, condensed heterochromatin structure? Here, we reveal surprisingly dynamic properties of interstitial heterochromatin in ESC and find well-known hallmarks of heterochromatin are compatible with rapid exchange of core histones while maintaining DNA in an inaccessible state.

## Results

**Interstitial heterochromatin in pluripotent stem cells.** Several studies have assessed genome-wide H3.3 dynamics in ESC, either measuring incorporation or turnover of ectopically tagged H3.3[5,8,9,34], albeit exclusively looking at dynamics related to enhancers and transcription. A recent study assessed nucleosome turnover using a method termed 'time-ChIP' in mouse embryonic stem cells as well as neural stem cells (NSC)[9], reporting high nucleosome turnover at enhancers correlating with DNA accessibility. For the time-ChIP method, SNAP-tagged H3.3 was expressed from a Tet-controlled transgene and biotin-labeled using the SNAP-tag at the onset of a time course (0 h). The biotin-labeled fraction then was observed over a time range of 3–12 hours (Fig. 1a, b). Dynamic regions are marked by H3.3 at 0 h and gradually decay toward the genome-wide average, implying replication-independent turnover of H3.3[9].

We reanalyzed time-ChIP datasets with respect to the known heterochromatic enrichment of H3.3. H3.3-enriched regions in ESC called from the initial time point (0 h) overlapped well with those determined for endogenous histone H3.3[21], including those peaks sharing H3.3 and H3K9me3 (H3.3 + H3K9me3) (Fig. 1c, d). As an example, H3.3 enrichment tracked well with known ESC enhancers and H3K27ac marks, but to similar levels with a nearby IAP ERV (Fig. 1b and Supplementary Fig. 1a). NSC showed globally similar turnover kinetics (Fig. 1a and Supplementary Fig. 1b, c) and maintained many of the H3.3 peaks defined in ESC (Fig. 1c, d). However, among those H3.3 peaks specifically lost in NSC were essentially all H3.3 + H3K9me3 peaks (Fig. 1c, d), as exemplified by an IAP ERV localized in the Hist1 histone gene cluster (Supplementary Fig. 1b). Thus, as previously concluded[21], co-enrichment of H3.3 and H3K9me3 is a feature of pluripotent stem cells, which appears to be lost early upon differentiation. Interestingly, H3K9me3 and H3.3 have also been studied genome-wide during reprogramming of mouse fibroblasts to induced-pluripotent cells[35]; while H3K9me3 becomes enriched over IAP ERV in pre-iPSC, H3.3 is absent at this stage (Supplementary Fig. 1d). Pre-iPSC are stable late-stage reprogramming intermediates that can be converted to the pluripotent state through MEK inhibition. Thus, these data corroborate a link between pluripotency and the acquisition of H3.3 at interstitial heterochromatin, where H3K9me3 has already been established during reprogramming.

Annotating H3.3 peaks with a 15-state ChromHMM model generated with seven histone modifications and RNA Polymerase II (RNAP2) profiles[36,37], we found that H3.3-only peaks overlapped largely with active transcription (states 1–3) and enhancers (states 4–9) (Fig. 1d and Supplementary Fig. 2a). H3.3 + H3K9me3 peaks were assigned to heterochromatin (states 12–14) (Fig. 1d and Supplementary Fig. 2a), defined by the absence of H3K4me3, H3K4me1, H3K27ac, H3K36me3, and RNAP2[37], in line with the fact that H3K9me3 appears mutually exclusive with these signatures of active chromatin. Binning the genome into 5 kb windows, we found a bimodal H3.3 distribution, with regions of very high and very low levels of H3K9me3 coinciding with the highest H3.3 enrichment (Fig. 1e). Overlaying the 5 kb bins with repeat annotation, we confirmed that the strongest H3.3 + H3K9me3 co-enrichment was linked to the presence of an IAP element (Fig. 1e and Supplementary Fig. 3). Further, ETn and MusD elements also defined a subset of H3.3 + H3K9me3 co-enriched regions (Supplementary Fig. 3). It is known that a small fraction of ETn/MusD elements are active and, in fact, highly transcribed. This is evident from elongating RNA Polymerase II running into the 3′ flanking region of few elements, and the complete absence of H3K9me3 around these elements (Supplementary Fig. 4a, b). Since we cannot

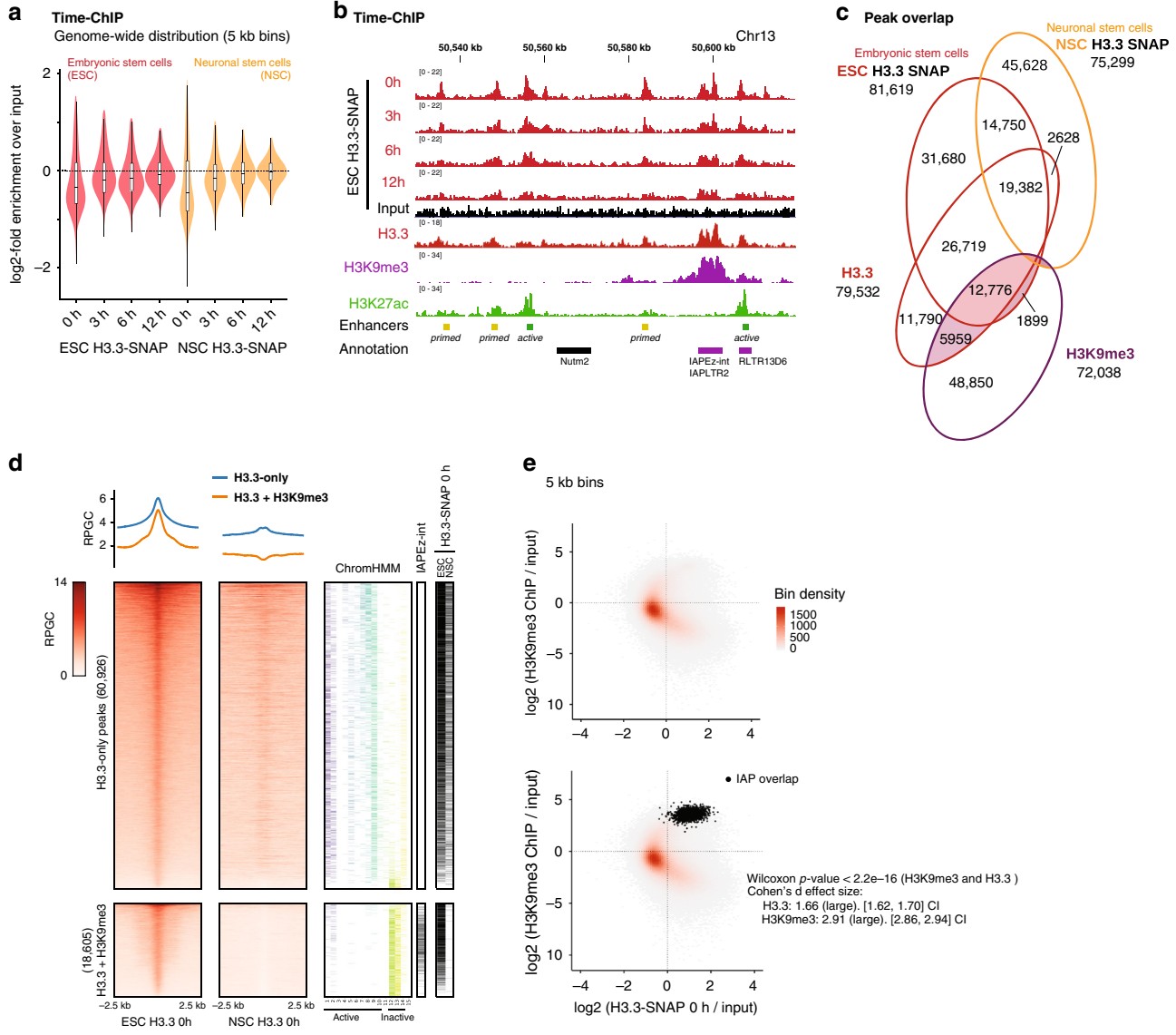

**Fig. 1 Ectopic H3.3-SNAP shows ESC-specific enrichment at interstitial heterochromatin. a** Violin plots showing genome-wide distribution of biotin-labeled H3.3-SNAP[9] in 5 kb windows, in mouse embryonic stem cells (ESC) and neuronal stem cells (NSC). Biotin labeling of H3.3-SNAP was performed at 0 h and followed over subsequent 3, 6, and 12 h[9]. Each datapoint is calculated as the log2 value of the mean coverage in each sample bin divided by the mean coverage value of its corresponding input sample bin. Data shown from *n* = 1 biological replicate. Source data are provided as a Source data file. **b** Genome tracks showing example region of time-ChIP[9] in ESC showing overlap of H3.3-SNAP and endogenous H3.3[21] with H3K27ac[20] at enhancers and H3K9me3[71] at IAP ERVs. All tracks were normalized to Reads Per Genomic Content (RPGC). See also Supplementary Fig. 1 for controls and additional tracks. **c** Venn diagrams showing overlap of H3.3-SNAP 0 h[9] peaks with endogenous H3.3 peaks in ESC[21], H3K9me3[71], and H3.3-SNAP 0 h peaks in NSC[9]. **d** Read density heatmaps and average profiles of H3.3-SNAP in ESC and NSC, at ESC H3.3 peaks[21]. Peaks were separated in two categories, H3.3-only and H3.3 + H3K9me3, according to coincidence with H3K9me3 peaks[21]. Overlap with 15 ChromHMM states[37], IAP ERV elements (from RepeatMasker) and H3.3-SNAP peaks in ESC and NSC is shown as additional heatmaps where the coincidence with a chromatin state or peak from the respective annotation set is indicated with a colored or black line. **e** Scatter plot showing relationship between H3.3[21] and H3K9me3[71] assessed over 5 kb bins genome-wide. Color scale represents density of underlying data points (5 kb bins). Bottom graph shows bins overlapping with IAP ERVs in black. Two-sided Wilcoxon rank sum ($p < 2.2e^{-16}$) and Cohen's effect size d (H3.3: 1.66 +/−0.04; H3K9me3: 2.91 ± 0.05) tests show significant enrichment for both at H3.3 and H3K9me3 in these 5 kb bins. Overlap with additional repeat families are shown in Supplementary Fig. 3. Source data are provided as a Source data file.

unambiguously attribute ChIP-Seq reads within the highly conserved ETn/MusD internal sequence to specific instances, we are not able to discern how much H3.3 incorporation can be attributed to the small fraction of active versus majority of repressed copies. Therefore, we did not further focus on ETn/MusD elements. Amongst the IAP repeat family, none of the annotated instances showed similar evidence of transcriptional activity (Supplementary Fig. 4c, d), and RNA-Seq transcript levels

were 1–2 orders of magnitude lower than those of ETn (Supplementary Fig. 4e).

**Rapid histone turnover in the absence of DNA accessibility.** Using the time-ChIP data, we next assessed the turnover of labeled histone H3.3 at H3.3-only and H3.3 + H3K9me3 peaks. Decay of H3.3 toward background levels over the 12 h chase

period was observed at all peaks irrespective of their H3K9me3 status (Fig. 2a). Turnover at H3.3 + H3K9me3 enriched regions was also observed in another Tet-OFF dataset[5] (Supplementary Fig. 5a). A different dataset, applying a pulse of tagged histone H3.3[34], showed that newly synthesized histone H3.3 is rapidly incorporated at both H3.3-only and H3.3 +

H3K9me3 peaks (Fig. 2b and Supplementary Fig. 1a, 5). An orthogonal method to measure nucleosomal H3-H4 incorporation and turnover, CATCH-IT, relies on the metabolic labeling of newly synthesized proteins with Azido-homoalanine (AHA)[38]. CATCH-IT, previously performed in mouse ESC[6], confirms the dynamic exchange of core histones at both H3.3-only and H3.3

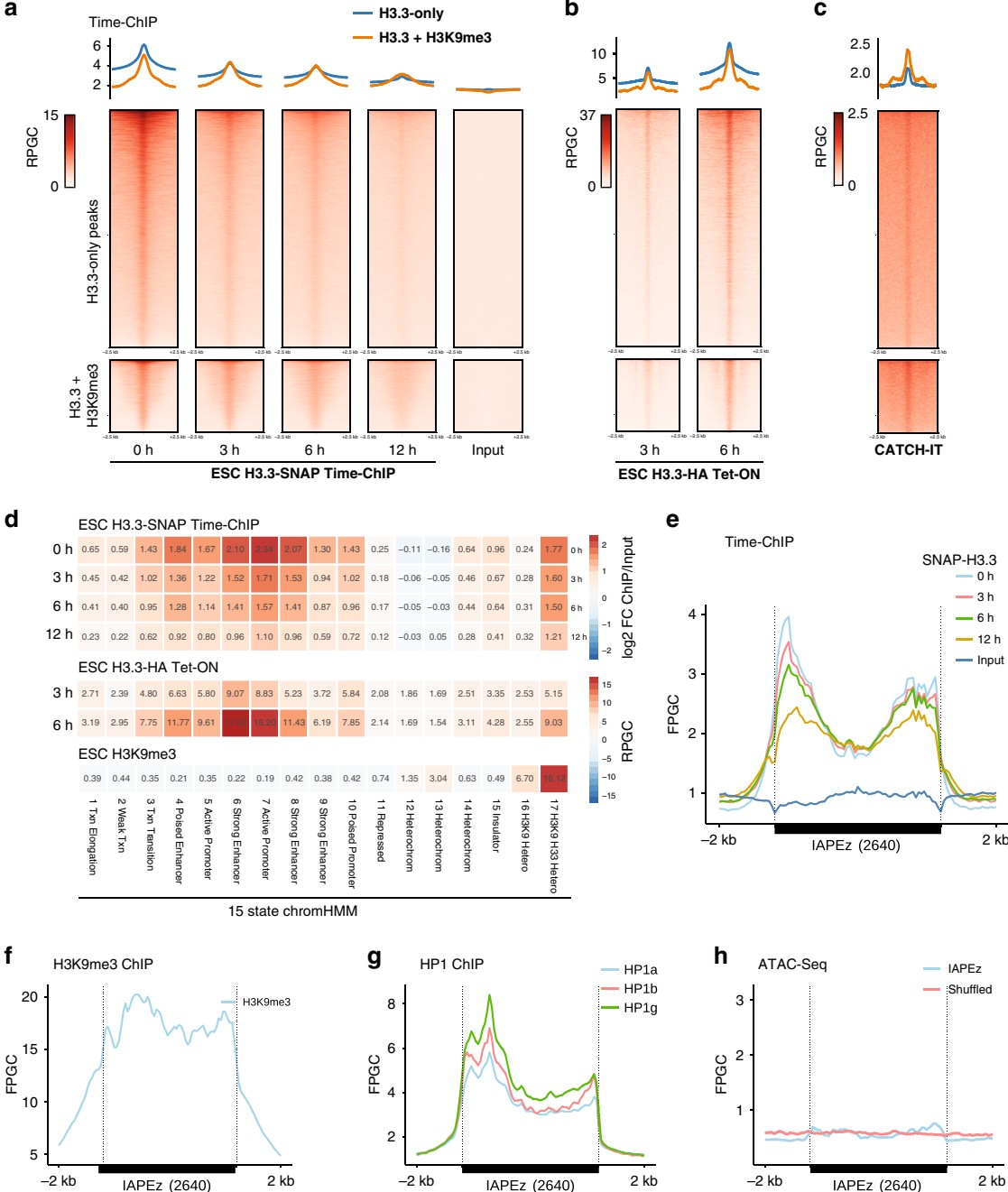

**Fig. 2 Rapid histone turnover at interstitial heterochromatin in the absence of DNA accessibility. a** Read density heatmaps and average profiles of H3.3-SNAP time-ChIP data[9] over two classes of H3.3 peaks, H3.3-only and H3.3 + H3K9me3 peaks. All data is normalized to Reads Per Genomic Content (RPGC). **b** Read density heatmaps and average profiles of H3.3-HA pulse data[34] over H3.3 peaks. **c** Read density heatmaps and average profiles of CATCH-IT data[6]. **d** Mean read density heatmap showing enrichments of H3.3-SNAP time-ChIP (log2-fold change over input) over 15 ChromHMM states[37] as well as H3K9me3 and H3.3 + H3K9me3 enriched regions[21]. Additional replicates are shown in Supplementary Fig. 5b. For enrichment analysis by repeat families see also Supplementary Fig. 6. Source data are provided as a Source Data file. **e** Average coverage of H3.3-SNAP time-ChIP over 2640 shared IAP ERVs. Fragments defined by paired-end reads were piled up and normalized by 1x Genome coverage (Fragments Per Genomic Content, FPGC). See Supplementary Fig. 5c for coverage of uniquely mappable reads only. **f** Average coverage (FPGC) of H3K9me3 ChIP[71] over 2640 shared IAP ERVs. **g** Average coverage (FPGC) of BioChIP for HP1 isoforms[72] over 2640 shared IAP ERVs. **h** Average profiles of DNA accessibility[20] over 2640 shared ERVs, and 2640 random (shuffled) genomic regions of matching size. Source data are provided as a Source Data file.

+ H3K9me3 peaks (Fig. 2c). Crucial for this method, all non-histone proteins and H2A/H2B dimers are biochemically stripped of chromatin before performing the ChIP, revealing the incorporation dynamics of newly synthesized H3-H4 units irrespective of the H3 variant[38]. Thus, three independent datasets provide complementary evidence for nucleosome turnover at interstitial heterochromatin. While rapid H3.3 turnover has been attributed to "hyperdynamic" promoter and enhancer nucleosomes[8,9], our analysis suggests that similar dynamics also apply to H3.3 + H3K9me3 peaks.

We next compared H3.3 dynamics across 15 ChromHMM states[37] and H3.3 + H3K9me3 heterochromatic regions and found that the highest histone H3.3 turnover is observed at active promoters and enhancers followed by H3.3 + H3K9me3 regions (Fig. 2d and Supplementary Fig. 5b). Analyzing individual repeat families using UCSC RepeatMasker annotation, the ~340 bp IAPLTR1/1a long terminal repeats showed comparable turnover to active promoters and enhancers, followed by IAPLTR2 and internal IAP regions (Supplementary Fig. 6). Examining H3.3 dynamics across full-length IAPLTR1 elements, it was interesting to note that histone turnover was not confined to individually positioned nucleosomes but broadly occurred over LTRs and adjacent ~2 kb of internal region (Fig. 2e). Remarkably, regions of highest turnover were also highly enriched for H3K9me3, concomitant with HP1 binding (Fig. 2f–g). Prior analysis suggested that DNA accessibility at ERVs is generally low[29], in line with stereotypical heterochromatin properties. Our reanalysis of recent ATAC-Seq data in mESC[20] confirmed that IAPLTRs neither show defined nucleosome-free regions nor broader domains of accessible DNA (Fig. 2h). In summary, we find that histone turnover does not require or induce an increase in DNA accessibility at IAP ERVs (Fig. 2f), Instead, it appears that histone H3.3-H4 is swapped into chromatin directly replacing an existing H3-H4 histone dimer or tetramer, hinting at the possibility of a concerted mechanism.

**Gain in DNA accessibility in the absence of histone H3.3.** To achieve an exchange of core histones without rendering DNA at least transiently accessible, a tightly regulated supply of H3.3 would appear necessary. We thus wondered how a lack of H3.3 would affect DNA accessibility at H3.3 + H3K9me3 regions. ATAC-Seq profiles of wildtype and H3.3 knockout cells have previously been compared[20]. The authors note that DNA accessibility is largely unchanged across the genome in the absence of H3.3, including sites of rapid histone exchange, such as promoters and enhancers[20]. Loss of H3.3 was thus fully compensated at promoters and enhancers, presumably by assembly or recycling pathways for H3.1/H3.2. In line with the original report, our reanalysis of the ATAC-Seq signals at H3.3-only peaks showed that DNA accessibility was not altered in H3.3KO cells (Fig. 3a, b). Genome-wide analysis of 5 kb bins and ChromHMM regions provided no evidence of systematic changes in DNA accessibility (Fig. 3c and Supplementary Fig. 7), suggesting that by large, loss of H3.3 had no effect on the packaging of the genome.

On the basis of this, it was striking to find a roughly threefold increase in DNA accessibility in H3.3KO cells specifically at H3.3 + H3K9me3 peaks (Fig. 3a, b). Genome-wide analysis of 5 kb bins further confirmed in an unbiased manner that H3K9me3 + H3.3 co-enrichment regions showed the highest fold-increase in DNA accessibility in the absence of H3.3 (Fig. 3d and Supplementary Fig. 7). This suggested that H3.3, while dispensable for proper packaging of other regions of the genome, is required for maintaining a closed chromatin state at H3.3 + H3K9me3 heterochromatin.

As discussed above, a large fraction of H3.3 + H3K9me3 heterochromatin is contributed by IAP ERVs (Fig. 1e). Most IAP ERV containing 5 kb bins moved from a low DNA accessibility (roughly genome-average) to the top quartile of accessible regions genome-wide in H3.3KO cells (Fig. 3c). Although an increase in DNA accessibility could be indicative of a gain of euchromatic histone modifications, loss of H3.3 did not lead to a gain in histone H3K18, H3K27, H3K64 or H3K122 acetylation at H3.3 + H3K9me3 peaks (Fig. 3b and Supplementary Fig. 8). This is in line with prior observations that loss of H3.3 leads to only a partial loss of H3K9me3 modifications and a moderate increase in transcriptional activity[21,39].

We next wondered if specific functional DNA elements of the IAP ERV would become accessible in the absence of H3.3. In contrast to the typically narrow peaks over promoters and enhancers corresponding to transcription factor binding sites, ATAC-Seq signal increased broadly across LTR and ~2 kb of internal region of IAP ERVs, in a profile that closely matched the distribution of H3.3 in wildtype cells (Fig. 3e and see Fig. 2e for comparison). Thus, H3.3 appears to play a specific role at IAP ERV in maintaining DNA in a tightly packaged and inaccessible state. Analysis of uniquely mappable paired-end reads within and adjacent to IAP ERVs supports a widespread effect on many individual instances of the IAP ERV family (Fig. 3f and Supplementary Figs. 7e and 9). In addition to full-length or truncated IAP ERVs, the mouse genome contains many single LTRs (predominantly of the IAPLTR2 family). Using only uniquely mappable read pairs, we found that the 651 H3.3-enriched orphan LTRs also gained DNA accessibility in the absence of H3.3 (Fig. 3g).

In summary, we find that interstitial heterochromatin maintains underlying DNA in an inaccessible state, a key characteristic of all heterochromatin. Despite this, interstitial heterochromatin is permissive to nucleosome turnover and histone H3.3 incorporation, providing a mechanistic explanation for the co-occurence of H3.3 and H3K9me3. According to the collective evidence from the data analyzed here, nucleosome eviction and assembly of new histone H3.3 nucleosomes must be accomplished through a mechanism that does not allow the underlying DNA to become accessible in the process. Loss of histone H3.3 may uncouple eviction of existing nucleosomes from immediate reassembly of a new nucleosome and consequently lead to the observed broad gain in DNA accessibility.

**Strict requirement for histone H3.3 nucleosome assembly.** Histone H3.3 is known to be deposited at interstitial heterochromatin by the histone chaperone DAXX[21,40–42]. DAXX has been proposed to contribute to ERV silencing through its histone chaperone activity and, independently, recruitment of histone deacetylases to ERVs[21,39]. Loss of H3.3 also reduces global DAXX levels in mouse ESC, but H3.3 + H3K9me3 regions did not gain histone H3 acetylation in the absence of H3.3 (Supplementary Fig. 8). This suggests that impaired nucleosome assembly in the absence of H3.3 was the cause of the observed opening of chromatin (Fig. 3). To test if the role of H3.3 was directly linked to nucleosome assembly at ERVs, we generated a panel of rescue cell lines based on the H3.3KO cell line. We either stably expressed histone H3.3, the replication-dependent histone H3.2, or an H3.3 with mutations L126A I130A previously shown to inhibit stable formation of nucleosomes[39,43] (Fig. 4a). We performed ATAC-Seq on these cell lines and observed that restoration of H3.3 protein specifically reverted chromatin accessibility at IAP ERV regions, (Fig. 4b–d and Supplementary Fig. 10a), almost to the low levels of wildtype H3.3 (Fig. 4c). Expression of replication-dependent H3.2 did not rescue the heterochromatin phenotype,

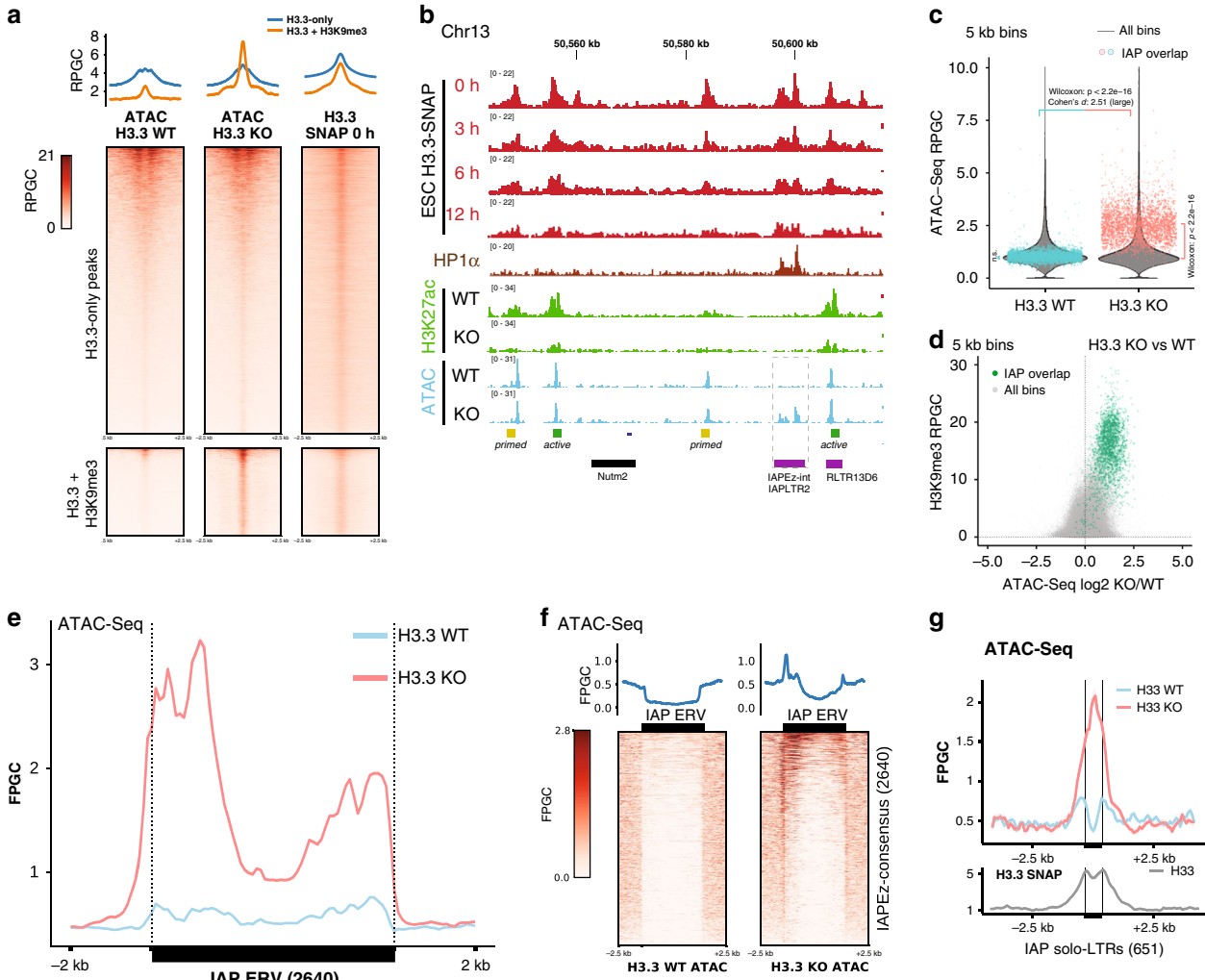

**Fig. 3 Interstitial heterochromatin becomes accessible in the absence of H3.3. a** Read density heatmaps and average profiles of H3.3-SNAP time-ChIP[9] and ATAC-Seq[20] over two classes of H3.3 peaks, H3.3-only and H3.3 + H3K9me3 peaks. See Supplementary Figs. 7 and 8 for additional analysis. **b** Genome tracks showing H3.3-SNAP time-ChIP[9], H3K27ac[20], and ATAC-Seq[20] signal over example region on Chr13. All tracks are normalized to Reads Per Genomic Content (RPGC). Source data are provided as a Source Data file. **c** Violin plots showing genome-wide distribution of ATAC-Seq signal in 5 kb bins, in H3.3 wildtype (WT) and knockout (KO) cell lines. Overlayed in color are 2797 individual bins overlapping with 2640 shared IAP ERVs. Two-sided Wilcoxon rank test ($p < 2.2e-16$) and large Cohen's effect size ($d = 2.51 \pm 0.07$) show significant increase in accessibility for IAP ERV containing bins in H3.3 KO. Genome-wide comparison between H3.3 WT and KO shows a significant ($p = 6.4e-14$) difference but negligible effect size ($d = 9.4e-6$). IAP-overlapping bins are not different from genome-wide average in H3.3 WT ($p = 0.97$), but significantly different ($p < 2.2e-16$) in H3.3 KO. Data shown from $n = 1$ biological replicate. Source data are provided as a Source data file. **d** Scatter plot of 5 kb bins, showing H3K9me3 level versus log2-fold change in ATAC-Seq signal upon H3.3 knockout. Bins overlapping with IAP ERVs are drawn in green. **e** Average coverage of ATAC-Seq signal over 2640 shared IAP ERVs in H3.3 wildtype (WT) and knockout (KO) cell lines. Fragments defined by paired-end reads were piled up and normalized to 1x Genome coverage (Fragments Per Genomic Content, FPGC). **f** Density heatmaps and average profiles of ATAC-Seq signal using only uniquely mappable reads, over 2640 shared IAP ERVs and flanking regions. See Supplementary Fig. 7e for quantitative analysis and Supplementary Fig. 9 for further comparison with H3.3 and H3K9me3 ChIP profiles. **g** Average coverage (FPGC) of ATAC-Seq signal (top) and H3.3-SNAP ChIP using only uniquely mappable reads, over 651 IAP solo LTRs (mostly IAPLTR2). Source data are provided as a Source data file.

and neither did the H3.3 L126A I130A mutant (Fig. 4b, c). The effect was also apparent in the 5′ flanking region of IAP ERVs and internal regions counting only uniquely mappable reads (Supplementary Fig. 10a) and no difference was observed at random control regions (Supplementary Fig. 10b). Together these data show that H3.3 is indeed required as a substrate for new nucleosomes to form at heterochromatic regions.

**Smarcad1 evicts nucleosomes at heterochromatin.** To explain the need for replication-independent nucleosome assembly, we sought to elucidate how existing nucleosomes could in the first place be evicted from a region that is highly bound by HP1 proteins (Fig. 4d) and generally considered compacted and repressed. Importantly, recent structural studies of the HP1-bound dinucleosome corroborate the view that heterochromatin can be accessible for nucleosome remodeling[32]. In principle, two possible mechanistic models could be imagined: either, a concerted mechanism replaces an existing H3-H4 dimer or tetramer in the nucleosome with a H3.3-H4 variant in a single step, or the two steps are uncoupled, essentially first creating a nucleosome-free region that is subsequently recognized by DAXX to deposit H3.3-H4. The former would be akin established action of Swr1 and INO80 chromatin remodeling complexes exchanging H2A to

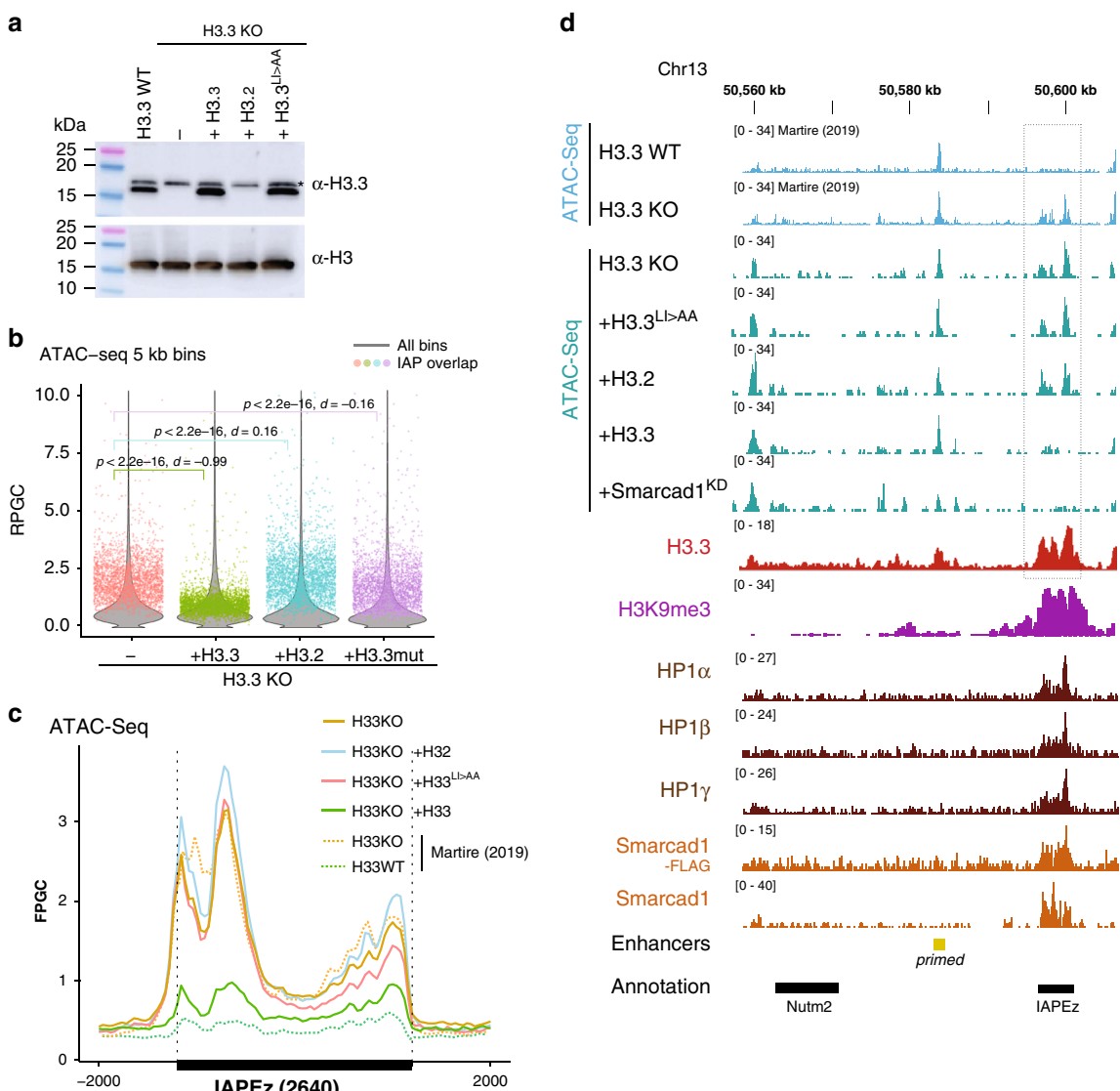

**Fig. 4 Histone H3.3 nucleosome assembly is strictly required for maintaining inaccessible state. a** Western blot showing complementation of H3.3 knockout (KO) cell line with transgenes stably expressing wildtype H3.3, H2.2 or a H3.3 L126A I130A (H3.3[LI>AA]) mutant incapable of forming nucleosomes. Representative image of $n = 2$ independent experiments. **b** Violin plots showing genome-wide distribution of ATAC-Seq signal in 5 kb bins, in H3.3 KO cell line, or H3.3 KO cell lines complemented with wildtype H3.3, H3.2 or a H3.3[LI>AA]. Overlayed in color are 2797 individual bins overlapping with 2640 shared IAP ERVs. Two-sided Wilcoxon rank test $p$ value and Cohen's effect size d are given comparing complemented cell lines with the parent H3.3 KO cell line. Data shown from $n = 1$ biological replicate. Source data are provided as a Source data file. **c** Average coverage of ATAC-Seq signal over 2640 shared IAP ERVs including data from Martire et al.[20], and this study. Fragments defined by paired-end reads were piled up and normalized to 1x Genome coverage (Fragments Per Genomic Content, FPGC). ATAC-Seq coverage in H3.3 KO is compared with H3.3 WT, and H3.3 KO complemented with wildtype H3.3, H3.2 or H3.3[LI>AA]. See Supplementary Fig. 10 for further coverage plots from unique reads. **d** Genome tracks showing ATAC-Seq signal, H3.3[21], H3K9me3[71], HP1a, HP1b, HP1g[72], and Smarcad1[47] over example region on Chr13. All tracks are normalized to Reads Per Genomic Content (RPGC).

H2A.Z dimers and vice versa[44,45]. The chromatin remodeler ATRX can form a complex with DAXX, but ATRX is not known to facilitate nucleosome eviction or exchange[40,42,46]. An uncoupled mechanism, on the other hand, would explain the observation that H3.3 depletion lead to an increase in DNA accessibility, thus the putative nucleosome eviction activity did not cease in the absence of H3.3. It caught our attention that the chromatin remodeler Smarcad1 has recently been linked to IAP ERVs in mouse ESC[47]. DAXX and Smarcad1 have been independently shown to interact with KAP1 but no biochemical or functional interaction between the two proteins has been reported[21,39,48]. However, Smarcad1 has further been shown to have nucleosome sliding and eviction activity[49,50], thus representing an interesting

candidate for mediating nucleosome turnover at ERVs. Reanalyzing Smarcad1 ChIP-Seq[47,51] showed that Smarcad1 specifically localized to H3.3 + H3K9me3 peaks, as previously observed for KAP1[21] (Fig. 5a, b). In addition to the focal enrichment at those peaks, Smarcad1 showed a similarly broad binding profile accross IAP ERVs as did histone H3.3 (Figs. 4d and 5c). This led us to hypothesize that Smarcad1 could be evicting nucleosomes across the entire ERV, generating free DNA followed by subsequent H3.3 deposition; Smarcad1-bound regions should thus show an increase in DNA accessibility in the absence of H3.3. We determined Smarcad1 binding sites from ChIP-Seq datasets[47,51] and found that a large fraction of Smarcad1 binding sites exhibited DNA-accessibility in the absence of H3.3 (Fig. 5d). They also

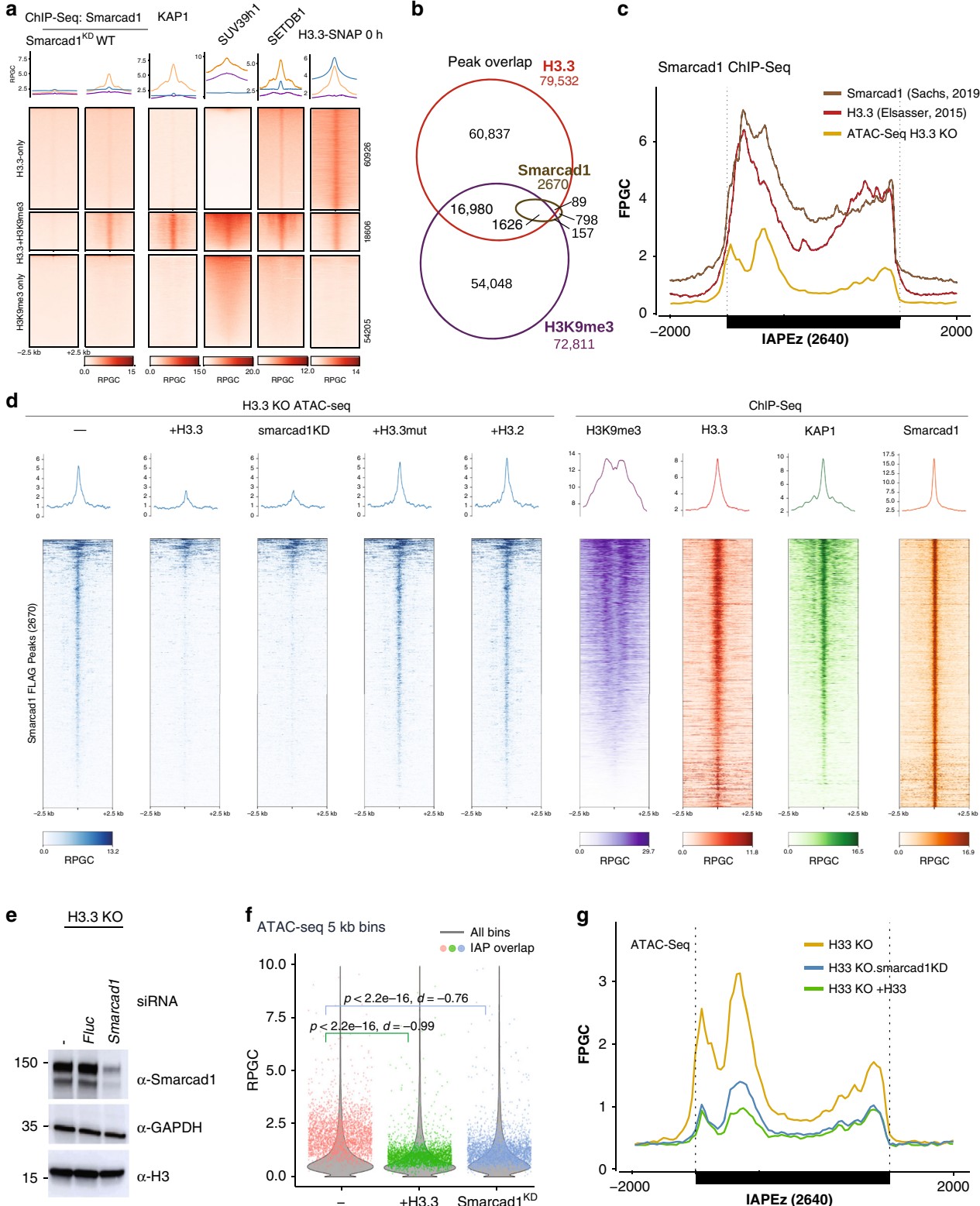

coincided with peaks of H3.3 and KAP1, while H3K9me3 levels are highest at adjacent regions (Fig. 5d). DNA accessibility was negligible in WT cells, and H3.3 expression in H3.3 KO cells, but not H3.2 or H3.3 L126A I130A mutant expression, restored inaccessible chromatin at Smarcad1 binding sites (Fig. 5d and Supplementary Fig. 11). In summary, Smarcad1 binding sites feature nucleosome turnover, H3.3 incorporation and a requirement for H3.3 to prevent DNA from becoming accessible.

To test if Smarcad1 activity involves the eviction of nucleosomes at heterochromatin sites as suggested by the above correlations, we knocked down *Smarcad1* by RNAi (Fig. 5e) in H3.3 KO cells. While *Smarcad1* knockdown did not completely abolish Smarcad1 protein (Fig. 5e), DNA accessibility at Smarcad1 binding sites was much reduced (Fig. 5d). This corroborated our hypothesis that Smarcad1 evicts nucleosomes at heterochromatin. Strikingly, *Smarcad1* knockdown closely

**Fig. 5 Smarcad1 chromatin remodeler evicts nucleosomes from interstitial heterochromatin. a** ChIP-Seq density heatmaps and average profiles of Smarcad1 in mouse ESC[47] including *Smarcad1* knockdown control, KAP1[71], SETDB1[73], Suv39h1[74], and H3.3-SNAP[9] over H3.3-only, H3K9me3-only and H3.3 + H3K9me3 peaks[21]. All data is normalized to Reads Per Genomic Content (RPGC). **b** Venn diagram showing overlap of Smarcad1 peaks (called from Smarcad1 and Smarcad1-FLAG ChIP-Seq[47]) with H3.3 and H3K9me3 peaks[21]. **c** Average coverage of Smarcad1[47] and H3.3[21] ChIP-Seq, as well as ATAC-Seq, over 2640 shared IAP ERVs. Fragments defined by paired-end reads were piled up and normalized to 1x Genome coverage (Fragments Per Genomic Content, FPGC). **d** ATAC-Seq and ChIP-Seq read density heatmaps and average profiles over 2670 Smarcad1 peaks. ATAC-Seq coverage in H3.3 KO cells and H3.3 KO cells complemented with wildtype H3.3, H3.2 or H3.3$^{LI>AA}$ is shown. Further, H3.3 KO cells were treated with siRNA for Smarcad1 (Smarcad1$^{KD}$). ChIP-Seq profiles for H3K9me3[71], H3.3[21], KAP1[71], Smarcad1[47] are shown. Additional control heatmaps are shown in Supplementary Fig. 11. **e** Western blot showing control and Smarcad1 siRNA treatment of H3.3 KO cell line. Representative image of $n = 3$ independent experiments. **f** Violin plots showing genome-wide distribution of ATAC-Seq signal in 5 kb bins, in H3.3 KO cell lines untreated, complemented with H3.3, or treated with Smarcad1 siRNA (Smarcad1$^{KD}$). Overlayed in color are 2797 individual bins overlapping with 2640 shared IAP ERVs. Two-sided Wilcoxon rank test *p* value and Cohen's effect size d are given as compared against an untreated H3.3 KO cell line. Data shown from $n = 1$ biological replicate. Source data are provided as a Source Data file. **g** ATAC-Seq average coverage (FPRC) over 2640 shared IAP ERVs for H3.3 KO cell lines untreated, complemented with H3.3, or treated with Smarcad1 siRNA. The second replicate of this experiment is shown in Supplementary Fig. 12.

mirrored the effect of reintroducing H3.3 in the H3.3 KO cell line with respect to DNA accessibility at IAP ERVs (Fig. 5d, f, g). Given this striking observation, we also knocked down *Smarcad1* in wildtype ESC, and in H3.3-rescue cell lines (Supplementary Fig. 12a). As expected from our original observation that DNA accessibility at IAP ERVs is already low in wildtype ESC, *Smarcad1* knockdown showed only a minor effect (Supplementary Fig. 12a). However, the small but significant decrease in accessibility observed when knocking down *Smarcad1* in either wildtype or H3.3-rescue cell lines (Supplementary Fig. 12b–e) agreed with our hypothesis that Smarcad1 nucleosome eviction activity leads to open chromatin, but is almost completely mitigated in wildtype cells by deposition of histone H3.3. We also performed *Atrx* knockdown, since ATRX is another chromatin remodeler that has been implicated in maintaining interstitial heterochromatin in a direct complex with DAXX and/or via its direct recognition of H3K9me3[39,40,42,52,53]. *Atrx* knockdown showed a small increase in accessibility over IAP ERVs (Supplementary Fig. 12b, e), suggesting that ATRX has an opposing activity to Smarcad1. Loss of ATRX has also been shown to reduce but not deplete histone H3.3 at IAP ERVs[21,52]. Together, these observations would be in line with ATRX being an accessory factor in assembling rather than evicting nucleosomes at IAP ERVs.

**H3.3 enrichment is a consequence of Smarcad1 activity.** Based on the ATAC-Seq experiments above we reasoned that Smarcad1 is a driver of dynamic heterochromatin and its activity both promotes and necessitates H3.3 deposition at interstitial heterochromatin. To test this hypothesis further, we performed H3.3 ChIP-Seq in wildtype ESC with or without *Smarcad1* knockdown. H3.3 was as expected highly enriched at IAP ERVs, but reduced by ~35% within the element and unique flanking regions upon *Smarcad1* knockdown. (Fig. 6a–c and Supplementary Fig. 13). Importantly, this effect was highly specific: H3.3-enriched promoters of highly expressed genes were not affected by *Smarcad1* knockdown (Fig. 6c); H3.3 + H3K9me3 peaks, which are co-occupied by Smarcad1 (Fig. 5a) peaks were significantly reduced upon *Smarcad1* knockdown, whereas H3.3-only peaks were unchanged (Fig. 6d). In fact, *Smarcad1* knockdown had an exquisitely specific effect on H3.3 density at H3K9me3-enriched regions and essentially did not affect any other functional region with known H3.3-enrichment, such as transcribed regions, active and poised enhancers or promoters (Fig. 6e, f). Analysis of a larger set of repetitive element families further confirmed that H3.3 was specifically lost from Smarcad1-occupied repeat elements (Supplementary Fig. 13c), and also non-repetitive inter-stitial heterochromatin as exemplified by a Smarcad1-occupied

20 kb heterochromatin island within the *Ezr* locus on chromo-some 17 (Supplementary Fig. 14).

Unlike histone H3.3, the H3K9me3 mark was only very weakly reduced by *Smarcad1* knockdown (Fig. 6d and Supplementary Fig. 13). Since SETDB1 is known to methylate nucleosomal H3 irrespective of the variant, this observation makes sense. According to our model, loss of Smarcad1 activity will stop nucleosome turnover, thus existing nucleosomes (including H3.3 nucleosomes) and their potential H3K9me3 mark are retained. Replication gradually dilutes H3.3 nucleosomes due to the incorporation of canonical H3.1/2, leading to the observed reduction of H3.3 (Fig. 6a), while H3K9me3 levels are presumably maintained by continuous SETDB1 activity.

In summary, the experiments above directly link H3.3 incorporation at interstitial heterochromatin to Smarcad1-mediated nucleosome eviction. Smarcad1 activity is unmasked in the absence of H3.3, then leading to measurable increase in DNA accessibility. The dependence on histone H3.3 implies that neither Smarcad1 nor other universal histone chaperones are proficient in reassembling nucleosomes using the evicted histones at these heterochromatic regions. Nucleosome eviction is a process inherently associated with histone acetylation and transcriptional activity[38,54,55]. Here we document for the first time nucleosome eviction at regions with stereotypical hetero-chromatin features. Continuous cycles of nucleosome eviction followed by de novo assembly of H3.3 nucleosomes characterize this unusual dynamic state of interstitial heterochromatin in mouse ESC.

## Discussion

Three key results presented above, (1) the dynamic incorporation and turnover of histone H3.3, (2) the opening of chromatin in the absence of H3.3, and (3) the suppression of chromatin opening by depleting Smarcad1, lead us to our final mechanistic model: Smarcad1 and H3.3 are opposing players at dynamic hetero-chromatin state, in mouse ESC (Fig. 7): Interstitial hetero-chromatin is established through master regulator TRIM28, which facilitates recruitment of many additional factors involved in heterochromatin formation, including methyltransferase SETDB1, Smarcad1, and DAXX. Smarcad1 activity evicts nucleosomes mainly consisting of canonical H3.1/2-H4 cores. These nucleosomes are not placed back after eviction, leaving transiently nucleosome-free regions. A supply of H3.3-H4 bound to DAXX serves as a substrate for immediate reassembly of nucleosomes, thus restoring the original compacted chromatin state. Therefore, initiated by Smarcad1, interstitial hetero-chromatin exists in a fluid state where nucleosomes are dyna-mically evicted and replaced. Since nucleosome reassembly cannot be achieved with canonical H3.1/2, but instead new

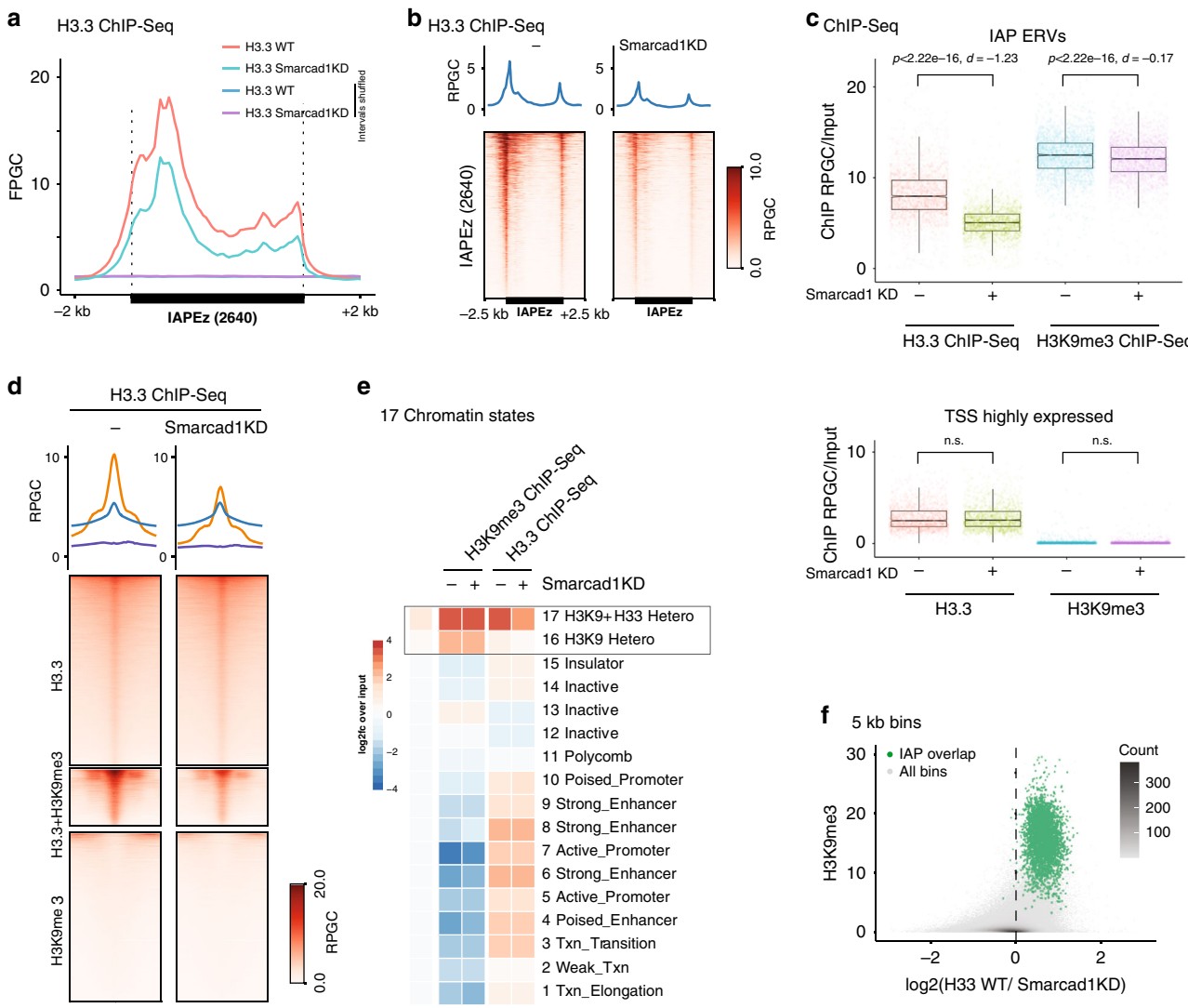

**Fig. 6 Smarcad1 drives nucleosome turnover and H3.3 incorporation at interstitial heterochromatin. a** Average coverage of H3.3 ChIP-Seq in H3.3 WT mESC, with or without 48 h *Smarcad1* knockdown, over IAP ERVs or random control regions. Fragments defined by paired-end reads were piled up and normalized to 1x Genome coverage (Fragments Per Genomic Content, FPGC). **b** Density heatmaps and average profiles of ATAC-Seq signal using only uniquely mappable reads, over 2640 shared IAP ERVs and flanking regions. **c** Boxplots showing H3.3 and H3K9me3 ChIP-Seq density at 2640 shared IAP ERVs (top) or 2024 transcription start sites (TSS) of highly transcribed genes (bottom). Tukey-style center (median), box (first and third quartiles) and whiskers (the maximal 1.5x IQR range). Two-sided Paired *T* test *p* value and Cohen's effect size d are given for each pairwise comparison. Data shown from *n* = 1 biological replicate. Source data are provided as a Source Data file. **d** ChIP-Seq density heatmaps and average profiles of H3.3 in H3.3 WT mESC with or without 48 h *Smarcad1* knockdown over H3.3-only, H3K9me3-only, and H3.3 + H3K9me3 peaks[21]. **e** Mean ChIP-Seq read density heatmap showing enrichments of Smarcad1-FLAG[47], H3K9me3, and H3.3 over 15 ChromHMM states[37] as well as H3K9me3 and H3.3 + H3K9me3 enriched regions[21]. Source data are provided as a Source data file. **f** Density scatter plot of 5 kb bins, showing H3K9me3 level versus log2-fold change in H3.3 enrichment upon *Smarcad1* knockdown. Bins overlapping with IAP ERVs are drawn in green. Source data are provided as a Source data file.

histone H3.3-H4 is incorporated into the chromatin fibre, our results imply a continuous need for SETDB1 to maintain high H3K9me3 levels. Despite these mechanistic insights, the purpose of such dynamic remodeling of heterochromatin remains unclear. While we did not observe a loss of H3K9me3 within the 48 h timeframe of *Smarcad1* knockdown, stable depletion of Smarcad1 has been shown to reduce H3K9me3 levels and KAP1 occupancy at IAP ERVs[21,47], concomitant with an increase in expression[21,47]. This suggests that the dynamic process reinforces rather than disrupts heterochromatin; transient opening of chromatin could allow KRAB-ZFP proteins to access the underlying DNA sequence and consequently amplify the repressive domain by recruitment of additional KRAB-ZFP/KAP1 corepressor complexes along the repetitive DNA sequence. Reassembly of

nucleosomes with the replication-independent substrate histone H3.3 would subsequently be necessary to maintain a compact chromatinized state with high levels of H3K9me3. Of note, our data does not exclude a role for additional chromatin remodelers in the process of eviction of reassembly of nucleosomes.

We have established that dynamic incorporation of H3.3 at heterochromatin is restricted to pluripotent stem cells (Supplementary Fig. 1d) raising the additional question if and how this mechanism is tied to pluripotency. One explanation could be that the overall dynamic properties (decompacted chromatin, high mobility of histones and chromatin-associated proteins)[56] intrinsic to pluripotent chromatin provide a challenge for the canonical heterochromatin machinery, and the mechanism uncovered here is an evolutionary solution to maintaining a silent

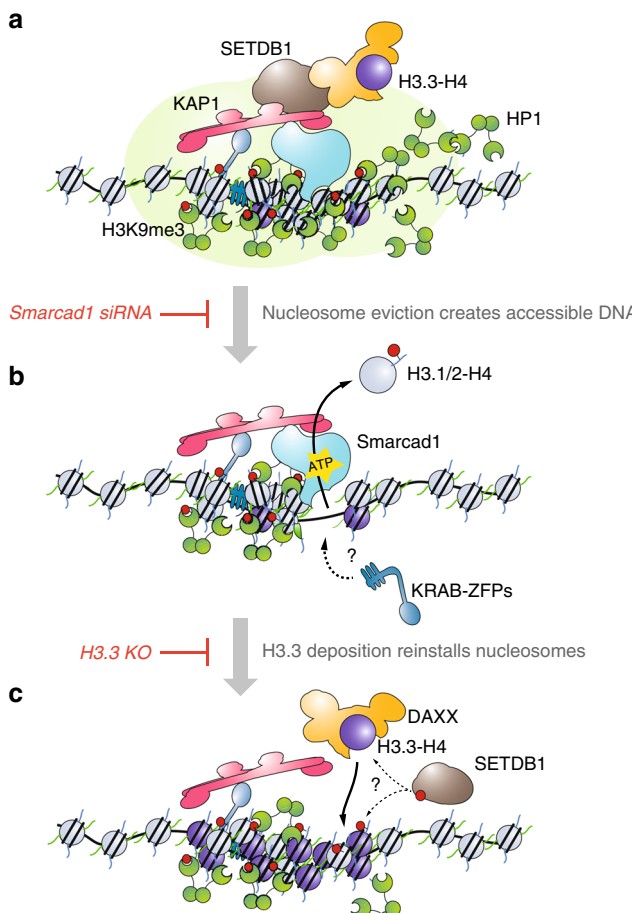

**Fig. 7 Molecular mechanism underlying dynamic heterochromatin.**
Proposed model for dynamic heterochromatin maintenance in mESC:
**a** KAP1/TRIM28 is recruited by DNA-binding KRAB-ZFPs to interstitial
repeats, such as IAP ERVs. KAP1/TRIM28 orchestrates heterochromatin
formation and maintenance, recruiting amongst others histone
methyltransferase SETDB1, histone H3.3 chaperone DAXX and chromatin
remodeler Smarcad1. Through SETDB1-mediated histone H3K9me3
methylation, HP1 is recruited and presumably contributes to
heterochromatin formation by compaction/phase separation. **b** Smarcad1
ATP-dependent remodeling activity leads to a disassembly of nucleosomes
and permanent eviction of existing histones. The primary reason for histone
eviction is unknown but may facilitate access of KRAB-ZFPs to the
underlying DNA sequence to amplify KAP1 recruitment. **c** In wildtype cells
DNA is only accessible very transiently since nucleosomes are immediately
reformed using H3.3, not canonical histone H3.1/2. The deposited H3.3
may be newly synthesized, thus does not carry H3K9me3 methylation.
Hence, SETDB1 needs to reestablish H3K9me3 on the dynamic nucleosome
substrates. In the absence of H3.3, nucleosome reassembly is impaired,
leaving DNA accessible. In the absence of Smarcad1, nucleosomes are not
evicted, alleviating the requirement for H3.3 as a substrate for new
nucleosomes and suspending nucleosome dynamics at heterochromatin.

state in a dynamic chromatin environment. Smarcad1 is highly
expressed in naïve pluripotent cells, and protein levels strikingly
decrease upon exit from pluripotency[47,51], and Smarcad1 appears
to be required to maintain naïve pluripotency[51]. Thus, main-
taining heterochromatin in a dynamic state may not just be the
consequence but in fact a requirement for pluripotency.

In the current literature, heterochromatin is invariably asso-
ciated with an inert chromatin packaging[33]. Dynamic histone
exchange is inherently viewed as a mechanism that opposes
heterochromatin formation[57]. Collectively, our analyses shed first
light into a dynamic heterochromatin state that defies the ste-
reotypical distinction of euchromatin and heterochromatin.

## Methods

**Primary data analysis pipeline for ATAC-Seq and ChIP-Seq datasets.** FASTQ
files were generated with Illumina's Bcl2Fastq tool (v2.20) or original FASTQ files
were downloaded from GEO and European Nucleotide Archive (ENA) databases
(see Extended Data Table 1 for a list of accession IDs). Depending on the deposited
source file format, paired-end or single-end reads were mapped to mm9 reference
genome using bowtie2 (v 2.3.5.1)[58] and samtools (v 1.10)[59] deduplicated using
Picard (v 2.20.4) (http://broadinstitute.github.io/picard/) MarkDuplicates with
default settings and filtered for blacklisted regions. Bigwig files were generated from
resulting deduplicated and filtered bam files using deepTools (v 3.1.0)[60], normal-
ized to 1x Genome Coverage (Reads Per Genomic Content) using --normal-
izeUsing RPGC --effectiveGenomeSize 2150570000. In 1x Genome Coverage
tracks, values below 1 represent a depletion below the genome average whereas
values above 1 indicate fold-enrichment above genome average. Unless stated
otherwise, downstream analysis was performed on the normalized bigWig files.
Where available, we evaluated all experimental replicates (see Supplementary
Fig. 5). For simplicity, figures show the first replicate.

**Track visualization.** Genomic tracks were visualized from 1x normalized bigWig
files using Integrated Genome Viewer (IGV, v2.3)[61]. Y axis corresponds to Reads
Per Genomic Content (RPGC).

**Peak calling.** Peaks were calculated using MACS2 (v 2.1.2)[62] callpeak using
parameters: broadPeak, default --broad-cutoff 0.05, corresponding input alignment
as control. The three replicates were combined to obtain a reliable set of peaks. A
peak is considered if it is called in at least two out of the three replicates. This was
performed for both ESC and NSC at initial timepoint (0 h). The region considered
as a reliable peak is then the result of merging the overlapping called peak regions.

**Bin-based analyses.** For the genome-wide coverage analyses in 5 kb bins, mean
coverage bin files were generated with deepTools multiBigWigSummary from the
final bigWig files using a bin size of 5 kb. Where available, input datasets were used
to normalize the ChIP-Seq coverage by calculating the log2 of the ChIP/input ratio.

**Repeat element annotation.** To filter the many instances of short and fragmented
repeat occurrences in the genome, we curated repeat elements from UCSC
RepeatMasker as follows: mm9 RepeatMasker track was curated to merge LTRs
and internal sequence into a single continuous interval. For this, adjacent elements
(allowing for no gaps) were clustered and merged with BEDTools (v2.27.1)[63] into
single elements, keeping the name of the largest fragment for the cluster (typically
the internal sequence, e.g. IAPEz-int, that defines the repeat family). Any single or
merged element shorter than 2 kb was removed, elements larger than 2 kb were
retained as continuous intervals and used as repeat annotation.

**Curated set of shared IAPEz elements across datasets from different mouse
strains.** A mm9 reference IAP ERV annotation was obtained from UCSC
RepMasker track, merging internal IAPEz-int and flanking LTR intervals and
retaining elements longer than 2 kb as described above (resulting in 3046 full-
length and truncated IAP ERVs). In each available paired-end ChIP-Seq and/or
Input dataset, read pairs overlapping with the reference IAP ERVs were retained if
at least one of the two reads was unambiguously assigned to one genomic match
(thus, the pair could be uniquely mapped with bowtie2 mapping quality MQ > 10).
One or more such anchoring read was used as evidence that the IAP ERV insertion
is present in the respective mouse strain. Evidence from all ChIP-Seq and matched
inputs were collectively used to define the IAP ERVs present in a given cell line/
mouse background. Finally, the annotation of 2640 shared IAP ERVs was obtained
by retaining only those IAP ERVs present in all the cell lines (see Supplementary
Fig. 15).

**Annotated bins.** Genome-wide 5 kb bins were intersected with our repeat anno-
tation using BEDTools intersect[63]. Bins were assigned the repetitive element group
of the element that overlapped the most with each bin, in the unusual case where
the same bin overlaps more than one repetitive element.

**Annotated peaks.** A peak list from endogenous H3.3 ChIP-seq[21] was intersected
with peaks called from Deaton et al.[9] as described above. Any amount of overlap >0
was considered. Other loci of interest were also intersected with these peaks in the
same fashion: IAPEz and ChromHMM15 regions. Since ChromHMM15 is a
partition of the genome, a peak is expected to fall in at least one category. For the
cases where a peak overlapped with more than one ChromHMM annotation, the
largest overlap was reported.

**Peak density heatmaps**. Values were calculated with deepTools (v 3.1.0) computeMatrix using parameters --referencePoint center and flanking regions of 2500 base pairs (-a 2500 -b 2500). In the cases where a different flanking region was used, such length is indicated in the plot. The resulting matrices were visualized using matplotlib (v 3.1.1) with the same parameter settings as deepTools plotHeatmap. Unless stated otherwise, the leftmost heatmap in a combined plot is sorted by mean coverage per locus and sorting order is applied to the rest of heatmaps in the same row. Profile plots on top of each heatmap are calculated as the mean values of the columns in the matrices generated by deepTools.

**IAPEz flanking H3.3 density heatmaps**. Values were calculated with deepTools. In this case the execution mode was reference-point with --regionBodyLength set to 6500. A set of uniquely mapped BigWig files was generated from the original alignment files using deepTools bamCoverage tool with parameters --minMappingQuality 10 --extendReads 150. For paired-end data, coverage was extended to match fragment length.

**Mean read density heatmaps**. Mean coverage values over each of the ChromHMM15 regions were calculated with R/rtracklayer library (v1.48.0)[64] and normalized to their corresponding input value (log2 ratio). Values obtained were visualized using R package pheatmap (v1.0.12).

**Statistics**. Plot statistics were calculated using R package ggpubr. Size effect estimations were calculated using R package effsize.

**Genome-wide enrichment over 5 kb bins for ES and NS cells at timepoints**. Bins were generated using deepTools multiBigWigSummary (see Bins). Each bin value was divided by its corresponding input value. Bins with zero values were excluded from this analysis. In addition, outliers were removed using Tukey's 1.5 IQR rule for the violin plots.

**Venn and Euler diagrams**. Venn diagrams were calculated and visualized using Intervene[65]. Proportional representations of these as Euler diagrams were visualized with R package eulerr.

**Profile plots**. Average plots over genomic regions were generated with ngsplot[66], including a modification to produce plots normalized to 1x Genome Coverage. The custom R code is provided as part of the public repository. Since ngsplot considers the coverage of actual fragments defined by paired-end reads for more realistic occupancy profiles, the *y* axis reports average Fragments Per Genome Coverage (FPGC) rather than RPKM.

**Cell culture**. Mouse embryonic stem cells (mESCs) were cultured feeder-free in 0.1% gelatin-coated flasks (Sigma, G1890) under standard conditions (10% $CO_2$, 5% $O_2$, 90% humidity, 37 °C) in KnockOut DMEM (LifeTechnologies, 10829018), 2 mM Alanyl-glutamine (Sigma, G8541), 0.1 mM non-essential amino acids (Sigma, M7145) 15% fetal bovine serum (FBS) (Sigma, F7524), 0.1 mM β-mercaptoethanol (Sigma, M3148), ESGRO Leukemia Inhibitory Factor (LIF) (Millipore, ESG1107), Penicillin-Streptomycin (Sigma, P4333) and 2i: 1 μM MEK inhibitor PD0325901 (Sigma, PZ0162) and 3 μM GSK3 inhibitor CHIR99021 (Sigma, SML1046). Generation of H3.3 Knock-Out has been previously described[6]. ESCs were routinely tested for mycoplasma.

**Cell lines**. H3.3 knockout (H3.3KO) and wildtype control (H3.3WT) cells were described previously[6,21]. Rescue cell lines expressing wildtype H3.3 (H3.3KO^+H3.3), H3.2 (H3.3KO^+H3.2) or H3.3 L126A I130A (H3.3KO^+H3.3LI−>AA) under EF1 promoter were generated by PiggyBac transgenesis. The coding sequences of the respective histones were cloned into a general-purpose PiggyBac vector containing an EF1 promoter, multiple cloning site, IRES and Blasticidin resistance gene[67]. 100 000 H3.3KO cells in single-cell suspension were transfected with 3 μL TransIT-X2 (Mirus, MIR 6004) with 1 μg total Plasmid DNA: PiggyBac targeting vector with histone gene of interest (GOI) and Super PiggyBac Transposase Plasmid pPBase (PBT) (PB200PA, System Bioscience Inc) at a ratio of 4:1, respectively, and plated in a 12-well plate. After 48 h, cells were split, allowed to recover overnight, and subsequently selected with 400–800 μg/ml Blasticidin (Invivogen, ant-bl-10p) for seven days. Experiments were performed at early passages and cell lines were routinely tested for mycoplasma contamination.

**siRNA transfection**. Approximately $5 \times 10^5$ ESCs in single-cell suspension in knockout medium without antibiotics were transfected with 10 uL Lipofectamine RNAiMax (Invitrogen, 13778030) and 100 nM siRNA [esiRNA SMARCAD1 (Sigma, EMU209081), esiRNA ATRX (Thermo, 187064/AM16704), esiRNA FLUC (Sigma, EHUFLUC)] and plated in six-well-plates for 48 h.

**ATAC-Seq**. Tn5 transposase was expressed and purified by Karolinska Institutet Protein Science Facility as described[68]. Tn5 transpososome assembly was

performed by mixing the following components: 0.125 volume of 200 μM Tn5ME-A/Tn5MErev, 0.125 volume of 200 μM Tn5ME-B/Tn5MErev, 0.4 volume of 100% glycerol, 0.12 volume of 2x dialysis buffer, 0.066 volume of 75.4 μM Tn5, 0.164 volume of nuclease-free water. ATAC-Seq in mouse embryonic stem cells was performed as described with minor modification[69]. Briefly, 100'000 cells were centrifuged at 600 g for 5 min at 4 °C. The pellets were resuspended in 200 μl lysis buffer (10 mM Tris-HCl pH 7.4, 10 mM NaCl, 3 mM $MgCl_2$, 0.1% NP40, 0.1% Tween-20, 0.01% digitonin) and incubated on ice for 3 min. After incubation, cells were centrifuged at 600 g for 3 min in the pre-chilled swinging-bucket rotor. The nuclei pellets were washed with wash buffer (10 mM Tris-HCl pH 7.4, 10 mM NaCl, 3 mM $MgCl_2$, 0.01% Digitonin). After washing, the nuclei were pelleted at 600 g for 3 min, resuspended in 50 μl transposase mixture with final concentration of 100 nM in-house Tn5 and incubated at 37 °C for 30 min with shaking. After transposition, the mixtures were digested with proteinase K at 63 °C for 1 h, followed by phenol-chloroform purification. Library PCR was performed with i5/i7 single index primers[69] and NEBNext Ultra II Q5 master mix (NEB, M0544S). The sequencing was performed on a Illumina Nextseq 500.

**Chromatin immunoprecipitation**. Cells were cross-linked with 1% methanol-free paraformaldehyde (PFA) in medium for 10 min at room temperature, quenched by adding Tris-HCl (7.8) to a final concentration of 0.75 M for 10 min at room temperature and washed twice with ice-cold PBS. 2x1e6 aliquots of the pelleted cells were resuspended in 50 μL cells Sonication Buffer 1 [50 mM Tris-HCl (pH 8.0), 0.5% SDS, 1x Protease Inhibitor Cocktail (PIC) (Sigma, 11873580001)] and chromatin fragmented by using a Bioruptor Plus sonication device (Diagenode) for 80 min (30 s on/30 s off, high) at 4 °C to ~150–300 bp. Samples were diluted further to a final volume 300 μL with Lysis Buffer [10 mM Tris-HCl (pH 8), 100 mM NaCl, 1% Triton X-100, 1 mM EDTA, 0.5 mM EGTA, 0.1% sodiumdeoxycholate, 0.5% N-laurolsarcodine, 1x PIC], centrifuged at maximum speed for 10 min at 4 °C and the cleared supernatants were used for chromatin immunoprecipitation (ChIP).

In all, 25 μl SureBeads Protein A (BioRad 161-4023) per IP were washed twice with PBS-T (PBS + 0.1% Tween 20) and coupled to one of the following antibodies in the same buffer for 1 h at room temperature with rotation: 4 μg a-H3.3 (Millipore, 09-838, Lot 3273632) or 2.5 μg a-H3K9me3 (Abcam, ab8898, Lot GR3244172-2). Beads were then washed quickly two times with PBS-T and one time with RIPA buffer [10 mM Tris-HCl (pH 8.0), 1% Triton X-100, 0.1% sodium deoxycholate, 0.1% SDS, 1 mM EDTA, 140 mM NaCl]. In all, 450 μL (3 × 1e6 cells) for a-H3.3 and 300 μL (2 × 1e6 cells) for a-H3K9me3 of the cleared lysate were added to the pre-coupled magnetic beads and ChIP assays were incubated further overnight at 4 °C with rotation. In all, 200,000 cells were saved as the input and processed through the remaining protocol in a manner similar to the IPs. Next, the beads were washed each time twice, 5 min each with 1 mL RIPA [10 mM Tris-HCl (pH 8.0), 1% Triton X-100, 0.1% sodium deoxycholate, 0.1% SDS, 1 mM EDTA, 140 mM NaCl], 1 mL RIPA high salt [10 mM Tris-HCl (pH 8.0), 1% Triton X-100, 0.1% sodium deoxycholate, 0.1% SDS, 1 mM EDTA, 360 mM NaCl], 1 mL LiCl Buffer [10 mM Tris-HCl (pH 8.0), 250 mM LiCl, 0.5% NP-40, 0.5% deoxycholate, 1 mM EDTA] and a quick wash with 1 mL TE buffer (Sigma, 93302), then resuspended in 100 μL ChIP elution buffer [10 mM Tris-HCl (pH 8.0), 1 mM EDTA, 0.1% SDS, and 300 mM NaCl] containing 0.25 mg/mL Proteinase K (Thermo, 25530015) and eluted overnight at 63 °C.

The DNA was isolated using 1x SPRI beads (Beckman Coulter). ChIP libraries were prepared using NEBNext Ultra II DNA library preparation kit (New England Biolabs, E7645) and amplified using NEBNext® Multiplex Oligos for Illumina (New England Biolabs, E7335) according to manufacturer's protocol. Quality assessment and concentration estimation of the purified DNA was done using the Qubit 3 (Life Technologies) and BioAnalyzer 2100 (Agilent). Each library was diluted to 4 nM and combined into a single pool before sequencing on the Illumina NextSeq500 platform (Illumina).

**Western blotting**. Cells were seeded in 6-well plates, harvested, washed with PBS, and resuspended in MNase Digestion Buffer [50 mM Hepes (pH 8), 1 mM $CaCl_2$, 0.2% NP-40, 1x PIC]. After lysis for 10 min on ice, each sample was incubated with micrococcal nuclease (MNase) (Thermo, 88216) 5 min at 37 °C. The MNase digestion reaction was quenched with EGTA and the lysates were further diluted with N- RIPA buffer for western blot [10 mM Tris (pH 8), 1 mM EDTA, 0.1% SDS, 0.1% sodium deoxycholate, 1% Triton X-100, 5% Glycerol, 140 mM NaCl, 1x PIC] and sonicated (3 cycles high intensity, 30 s on/off intervals). Samples were normalized based on protein concentration using BCA Protein Assay Kit (Life Technologies, 23227). 6x Laemmli sample buffer was added to the lysate and samples were incubated at 95 °C for 10 min. In all, 10–15 μL were loaded per lane on a 4–20% Mini-PROTEAN, TGX, Precast Protein Gels (Biorad). Proteins were transferred using Trans-Blot Turbo RTA Nitrocellulose Transfer Kit (Biorad), stained with Ponceau S (Sigma, P7170), blocked with 5% milk powder/TBS-Tween (Medicago, 097510100) and incubated with primary followed by secondary antibodies. Primary antibody was typically incubated overnight at 4 °C in 5% Milk/TBS-Tween whereas secondary was incubated for 1 h at room temperature. Horseradish peroxidase (HRP) signal was visualized using Luminata Forte (Millipore, WBLUF0500) substrate and ImageQuant LAS 500 (GE Healthcare). Uncropped source images for all blots are supplied in the Supplementary Information. Primary antibodies for western blotting: 1:200 a-ATRX (Santa Cruz, sc-

15408, Lot #A0915), 1:5000 a-H3 (Activemotif, 39763, Lot 20418023), 1:500 a-H3.3 (Millipore, 09-838 Lot 3273632), 1:5000 a-GAPDH (Millipore, AB2302, Lot 2967896), 1:2000 a-SMARCAD1 (Sigma, HPA016737, Lot 000004262). Secondary antibodies: 1:5000 a-Chicken-HRP (Invitrogen, A16054, Lot 58-39-081517), 1:5000 a-Mouse-HRP (BioRad, 1721011), 1:5000 a-Rabbit-HRP (BioRad, 172101).

**Oligonucleotides.** Oligonucleotides were ordered desalted from IDT. Sequences are given in Supplementary Table 1.

**Reporting summary.** Further information on research design is available in the Nature Research Reporting Summary linked to this article.

## Data availability

The data that support this study are available from the corresponding author upon reasonable request. The ATAC-Seq and ChIP-Seq data generated for this study have been deposited at the Gene Expression Omnibus under accession code GSE149080. Datasets from prior studies that were analyzed here: GSE78910, GSE57665, GSE123942, GSE97945, GSE48253, GSE59188, GSE42155, GSE114547, GSE114548, GSE63641, GSE90893, GSE73432, GSE57092 (see also Supplementary Data 1). Most of the downstream analyses come from bigWig files that can be obtained by running the Nextflow pipeline provided as part of the code repository attached to this publication. These bigWig files are also available upon request to the authors. Various metadata, genomic annotations used throughout the analyses, IGV sessions are available at the code repository and Mendeley Data rk28yn8gwg [https://doi.org/10.17632/rk28yn8gwg.2]. Source data are provided with this paper.

## Code availability

A public repository with the code used to analyze these datasets is available at: https://github.com/elsasserlab/publicchip. This includes a Nextflow[70] (version 19.07.0 build 5106—https://www.nextflow.io/) pipeline including primary analysis (download of the fastq files until BigWig generation), plus further code used for peak calling and aggregation, genome-wide bin generation, annotation, and visualization of the figures.

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

## Acknowledgements

Bioinformatics analyses were performed on resources provided by the Swedish National Infrastructure for Computing (SNIC) at Uppmax server (projects SNIC 2020/15-9, SNIC 2020/6-3, uppstore2018208, SNIC 2018/3-669, sllstore2017057, SNIC 2017/1-508). We thank the Protein Science Facility at the Department of Molecular Biochemistry and Biophysics at Karolinska Institutet for producing Tn5 transposase. We thank Johannes Heimgärtner for critical reading of the manuscript and members of the Elsässer lab for comments and help with experiments and analysis. S.J.E acknowledges funding by the Karolinska Institutet SFO for Molecular Biosciences, Vetenskapsrådet Junior Researcher Grant (2015-04815), H2020 ERC Starting Grant (715024 RAPID), Åke Wibergs Stiftelse (M15-0275), Cancerfonden (2015/430). J.L. acknowledges funding from the Chinese Scholarship Council. A.M.K. acknowledges funding from Barncancerfonden (TJ2016-0056).

## Author contributions

S.J.E. conceived study. C.N. performed all analyses. J.L. and A.M.K. performed ATAC-Seq and ChIP-Seq experiments. R.C. generated cell lines and helped with experiments. C.N. and S.J.E. generated figures and wrote the paper. All authors edited the paper.

## Funding

## Competing interests

The authors declare no competing interests.
