## [Peer Review File · Nature Communications]

Reviewers' comments:

Reviewer #1 (Remarks to the Author):

Although H3.3 histone variant is strongly associated with active transcription, a number of studies have shown its presence at repetitive genome including pericentric satellite DNA, telomere, retroviral DNA elements that are refractory to active transcription.

To investigate the role of H3.3 in heterochromatin maintenance, this study used a SNAP-tag based imaging system to investigate the turnover or genome-wide dynamics of H3.3 in mouse ES cells. The authors also investigated how the absence of H3.3 may affect DNA accessibility, both at the actively transcribed regions and at transcriptionally repressed heterochromatin. First, the authors found that the H3.3 mediated heterochromatin state was a feature linked to cellular pluripotency. Second, the H3.3+H3K9me3 (heterochromatin) regions and other H3.3 (euchromatin) sites showed a similar histone turnover dynamics. Thirdly, the loss of H3.3 led to increased DNA accessibility at H3.3+H3K9me3 heterochromatin regions. All of these observations and hypotheses proposed by the authors were interesting, however, there is a lack of experiments to test or address these hypotheses.

The manuscript can be improved if the following issues are addressed:

1) It is proposed that H3.3 specific heterochromatin state is a feature of pluripotent stem cells and that it is acquired during ips reprogramming. Is H3.3 important for the maintenance of cellular pluripotency? Is H3.3 K9me3 important for the maintenance of pluripotency? Do H3.3 knockout cells suffer from spontaneous differentiation? None of these questions was addressed.

5) What may be driving the changes in H3.3 K9me3 profiles during ES cell differentiation? Given that H3.3 deposition in the chromatin is not linked to replication, are the changes in H3.3 and K9me3 profiles during ES cell differentiation or ips cell reprogramming linked to the changes in replication timing?

3) It is surprising that there was no global change in DNA accessibility in H3.3 knockouts (except for H3.3-specific heterochromatin in H3.3 knockout). Could the authors explain this observation?

4) The highly dynamic H3.3 turnover at heterochromatin in ES cells is a very interesting observation. But, how is this possible? Is there a pathway that actively drives the turnover of H3.3 at the heterochromatin? The authors suggested a number of mechanisms in the 'Discussion' to explain how this may be possible, however, none of these mechanisms was tested in this study.

5) Do H3K9me3 sites (not H3.3 K9me3 sites) show a similar histone turnover dynamics in ES cells? What are the differences between H3.3K9me3 and H3K9me3 nucleosomes? Are H3.3K9me3-enriched nucleosomes more unstable compared to H3K9me3 nucleosomes?

Reviewer #2 (Remarks to the Author):

In this study, the authors examined the role of H3.3 in influencing nucleosome turnover at heterochromatic regions in mouse embryonic stem cells (ESCs). Combining time-ChIP, histone modification and ChIP-seq datasets, they show that nucleosome turnover at regions enriched with only H3.3 or H3.3+H3K9me3 occur at comparable time scales. Moreover, Intracisternal A Particle (IAP) retrotransposons are marked by H3.3, H3K9me3 and Hp1. At these sequences, the depletion of H3.3 results in change of chromatin accessibility as measured by ATAC-seq.

This paper presents a previously unappreciated aspect of how heterochromatic regions undergo

nucleosome turnover. The dynamics of this process and consequences are indeed quite interesting. The use of a number of sophisticated methods also serves to address the questions presented. However, I have several technical and conceptual concerns that I feel need to be thoroughly addressed. Below are my point-by-point comments:

Major Comments:

1. It has already been previously brought up in the field (Wolf et al. 2017 Nature) that when studying repetitive elements in mice, the genetic background is critical. Polymorphic insertions can lead to confounding genomic analysis results. In this study, it was not clear whether this was considered. The genetic background of some datasets like the time-CHIP from NSCs, Hp1 ChIP-seq, H3.3 WT and KO datasets should be addressed. A simple way to address this at least in part, would be to repeat some experiment in a C57Bl/6 or 129 line.
2. It was not clearly stated what proportion of the H3K9me3 marked regions are also H3.3 enriched. Are the loci considered the majority of K9me3 peaks or are they just a small proportion? Along the same lines, as Setdb1 is known to deposit H3K9me3 and repress many subfamilies of ERVs, it was not clear why IAPs were the main focus. For instance, families such as MusD/ETn, Gln and MMERVK10C all showed similar trends as IAP (Extended figure 3). It would be informative to at least consider those that have a substantial enrichment of H3.3.
3. With the analysis of ETn elements (Extended data figure 4), I have some concerns regarding the derived conclusions. ETn elements are far smaller and exist in fewer copies as IAPs, the qualitative comparison in panels A and B are not equivalent. It is unclear what the values of the heatmap are. It would be more informative to conduct a quantitative comparison on an individual elements level. Moreover, it was not explained clearly as to how the calculations of Extended data figure 4C was done. If copy number was not considered, then this would potentially create a bias. I don't doubt the result but the calculation is hard to understand. Also, it is worth noting that it has been long reported that some ETn/MusD elements have activity in ESCs and given that ETns are the non-autonomous versions of MusD, they are often considered together.
4. In figures 3E and Extended Figure 9, it looks as if the reduced mappability of the IAP internal sequences may be confounding the results. Given that uniquely mapped versus all mapped reads would give different results, it is difficult to conclude whether those that are losing H3.3 are actually gaining chromatin accessibility. Quantitative analysis on an individual element level would help. Also, consider showing the mappability of the IAP consensus sequence.
5. In addition to the issue of poor mappability, as the cells surveyed are asynchronous, I wonder if the change in chromatin accessibility is a feature of a heterogeneous population.
6. Generally, there is a lack of quantitative comparisons. Statistical tests can be applied to data presented in Figure 1A, 1E, 2C, etc.

Minor Comments

1. Consider including H3.3 ChIP-seq track for Figure 1B.
2. It's unclear what the 2 black and white columns for ESC/NSC H3.3-SNAP-0h show. The legends suggest they are peaks but I don't understand what data they present.
3. Extended figure 5b was not mentioned in text. I believe it corresponds to Extended Figure 6 on page 7.
4. Several figures are missing proper labelling of axis, legends, etc. (Figure 2E and 3F, Extended figures 1C, 4A-B, 5A and 8)
5. More details need to be provided in the methods and analysis. I was unable to find experimental

details as well as how some calculations were done. For instance, in Fig 1E, log2 over input is unclear to me. However, it's also not explained in the methods.

6. It's unclear why RPGC was used instead of typical RPKM.

7. The format of the manuscript appears to be different to others. The authors should break the results into sub-sections for easier reading.

Navarro et .al. response to reviewers comments

We thank the reviewers for their comments and have provided a response below. We have revised text and Figures 1-3 as well as Extended Data Figures in line with the reviewers comments. We have performed additional experiments, now presented in Figures 4 and 5. Our new data in conjunction with reanalysis of additional published datasets further extends the novelty of our study, identifying Smarcd1 as the source of nucleosome dynamics in heterochromatin: we first corroborated the role of histone H3.3 in maintaining a closed chromatin state by complementing H3.3 KO cells with H3.3, canonical histone H3.2 or a mutant H3.3 that cannot assemble nucleosomes. Only assembly-proficient H3.3 could revert the gain in DNA accessibility. This led us to search for a putative chromatin remodeler that could evict nucleosomes and we noticed a strong genome-wide correlation between Smarcd1 and heterochromatic H3.3. Indeed, knockdown of Smarcd1 reverted heterochromatin accessibility in H3.3 KO cells to a wildtype state, suggesting a new mechanism for nucleosome turnover and replenishment with histone H3.3 summarized in Figure 6:

Throughout the main text, new and major revised sections are indicated in blue.

ATAC-Seq data has been deposited at Gene Expression Omnibus under GSE149080.

```
To review GEO accession GSE149080:  
Go to https://www.ncbi.nlm.nih.gov/geo/query/acc.cgi?acc=GSE149080  
Enter token iraxsiqclduvlwr into the box
```

We have also made additional processed data and code available:

Code: <https://github.com/elsasserlab/publicchip>

BigWig files: <https://export.uppmx.uu.se/snic2020-6-3/Navarro/bw/>

Metadata: <https://export.uppmx.uu.se/snic2020-6-3/Navarro/doc/>

IGV Tracks: <https://export.uppmx.uu.se/snic2020-6-3/Navarro/igv/>

Bed files: <https://export.uppmx.uu.se/snic2020-6-3/Navarro/bed/>.

Reviewers' comments:

Reviewer #1 (Remarks to the Author):

Although H3.3 histone variant is strongly associated with active transcription, a number of studies have shown its presence at repetitive genome including pericentric satellite DNA, telomere, retroviral DNA elements that are refractory to active transcription.

To investigate the role of H3.3 in heterochromatin maintenance, this study used a SNAP-tag based imaging system to investigate the turnover or genome-wide dynamics of H3.3 in mouse ES cells. The authors also investigated how the absence of H3.3 may affect DNA accessibility, both at the actively transcribed regions and at transcriptionally repressed heterochromatin. First, the authors found that the H3.3 mediated heterochromatin state was a feature linked to cellular pluripotency. Second, the H3.3+H3K9me3 (heterochromatin) regions and other H3.3 (euchromatin) sites showed a similar histone turnover dynamics. Thirdly, the loss of H3.3 led to increased DNA accessibility at H3.3+H3K9me3 heterochromatin regions. All of these observations and hypotheses proposed by the authors were interesting, however, there is a lack of experiments to test or address these hypotheses.

The manuscript can be improved if the following issues are addressed:

1) It is proposed that H3.3 specific heterochromatin state is a feature of pluripotent stem cells and that it is acquired during ips reprogramming. Is H3.3 important for the maintenance of cellular pluripotency? Is H3.3 K9me3 important for the maintenance of pluripotency? Do H3.3 knockout cells suffer from spontaneous differentiation? None of these questions was addressed.

This question has already been addressed and previous publications reporting H3.3 knockdown or knockouts of H3.3 and is not the scope of this study. (Jang et al. 2015; Banaszynski et al. 2013; Elsässer et al. 2015; Martire et al. 2019; Gehre et al. 2020; Chronis et al. 2017; Fang et al. 2018). In short, H3.3 is dispensable for pluripotency (i.e. no increased spontaneous differentiation) but lack of H3.3 leads to differentiation defects and embryonic development cannot proceed without H3.3.

5) What may be driving the changes in H3.3 K9me3 profiles during ES cell differentiation? Given that H3.3 deposition in the chromatin is not linked to replication, are the changes in H3.3 and K9me3 profiles during ES cell differentiation or ips cell reprogramming linked to the changes in replication timing?

It is thought that the prevalence of H3K9me3 at transposable elements in pluripotent cells compensates for the fact that DNA methylation is neither necessary nor sufficient for their repression at this stage (Matsui et al. 2010; Rowe et al. 2010).

Replication timing of interspersed repeats such as ERVs and LINEs is controlled by the domain it is embedded in (insertions in early replicating regions are also replicated early). Thus, interstitial heterochromatin function, unlike broad heterochromatin at pericentromeric or peritelomeric regions cannot be controlled by replication timing.

3) It is surprising that there was no global change in DNA accessibility in H3.3 knockouts (except for H3.3-specific heterochromatin in H3.3 knockout). Could the authors explain this observation?

We absolutely agree that this is a surprising finding. The authors of the original study (Martire et al. 2019), see their Supplementary Figure 3 (<https://www.nature.com/articles/s41588-019-0428-5/figures/7>), have also noted this but did not speculate on the reasons.

We note that in many instances where histones are lost transiently, e.g. at promoters or enhancers, it appears possible to reassemble the nucleosome using the existing histone. Examples from transcription (FACT) and replication (MCM6/7, CAF-1) detail how histones are evicted, transiently stored on chaperones and then placed back. Even though H3.2/1 are predominantly incorporated during replication with the obligate need for the CAF-1 histone chaperone complex, it may be possible that in many instances 'old' H3.1/2 is used to reassemble nucleosomes if no H3.3 is available. Clearly, this is not possible at interstitial

heterochromatin according to our data, and it remains subject of our future studies if the ‘old’ histone is ever reused after eviction or even possibly degraded.

4) The highly dynamic H3.3 turnover at heterochromatin in ES cells is a very interesting observation. But, how is this possible? Is there a pathway that actively drives the turnover of H3.3 at the heterochromatin? The authors suggested a number of mechanisms in the ‘Discussion’ to explain how this may be possible, however, none of these mechanisms was tested in this study.

We have clearly been thinking along the same lines, and this has been our major focus in the revision. We now, through additional experiments presented in **Figure 4** and **5**, **Extended Data Figures 10, 11**, are able to answer this question and propose a mechanistic model in **Figure 6**: We identify Smarcd1 as the (or a dominant) factor for evicting nucleosomes at interstitial heterochromatin and driving force of chromatin opening in the ATAC assay. Through our own ATAC-Seq data we are able to show that H3.3 is required to reassemble nucleosomes in the wake of Smarcd1 eviction.

5) Do H3K9me3 sites (not H3.3 K9me3 sites) show a similar histone turnover dynamics in ES cells? What are the differences between H3.3K9me3 and H3K9me3 nucleosomes? Are H3.3K9me3-enriched nucleosomes more unstable compared to H3K9me3 nucleosomes?

Since H3K9me3-only peaks have no H3.3, there also cannot be H3.3 turnover. We have included this piece of information in New Figure 5a, as you can see, the profile for H3.3-SNAP is already flat at 0h, so there is not H3.3 incorporated or turned over.

Figure 5

CATCH-IT detects nucleosome turnover irrespective of the variant but does not provide evidence for dynamics at H3K9me3 peaks either:

Reviewer #2 (Remarks to the Author):

In this study, the authors examined the role of H3.3 in influencing nucleosome turnover at heterochromatic regions in mouse embryonic stem cells (ESCs). Combining time-ChIP, histone modification and ChIP-seq datasets, they show that nucleosome turnover at regions enriched with only H3.3 or H3.3+H3K9me3 occur at comparable time scales. Moreover, Intracisternal A Particle (IAP) retrotransposons are marked by H3.3, H3K9me3 and Hp1. At these sequences, the depletion of H3.3 results in change of chromatin accessibility as measured by ATAC-seq.

This paper presents a previously unappreciated aspect of how heterochromatic regions undergo nucleosome turnover. The dynamics of this process and consequences are indeed quite interesting. The use of a number of sophisticated methods also serves to address the questions presented. However, I have several technical and conceptual concerns that I feel need to be thoroughly addressed. Below are my point-by-point comments:

Major Comments:

1. It has already been previously brought up in the field (Wolf et al. 2017 Nature) that when studying repetitive elements in mice, the genetic background is critical. Polymorphic insertions can lead to confounding genomic analysis results. In this study, it was not clear whether this was considered. The genetic background of some datasets like the time-CHIP from NSCs, Hp1 ChIP-seq, H3.3 WT and KO datasets should be addressed. A simple way to address this at least in part, would be to repeat some experiment in a C57Bl/6 or 129 line.

This is an important concern and we have addressed this by performing all our analyses on a 'consensus' interval set which only considers IAP insertions present across the datasets from various sources and mouse backgrounds. The process is now explained in the Methods section and depicted in new **Extended Data Figure 12**, also reproduced below. Our strategy was to use paired-end sequencing data to verify insertion sites. We started with the IAP ERV insertions present in the mm9 reference genome and retained only those insertions that could be validated by uniquely mappable read pairs (typically one mate mapping in the LTR, the paired mate mapping to the unique flanking regions). Then we intersected the validated insertions across all the studies/cell lines and retained those 2640 insertions that were present in all datasets. The heatmaps of Extended Data Figure 4c using only uniquely mappable reads further adds evidence since H3K9me3 and H3.3 are present in the flanking regions of these 2640 IAP ERV insertions.

2. It was not clearly stated what proportion of the H3K9me3 marked regions are also H3.3 enriched. Are the loci considered the majority of K9me3 peaks or are they just a small proportion?

This is included now in Figure 1c, about 25% of the H3K9me3 peaks called overlap with H3.3.

Along the same lines, as Setdb1 is known to deposit H3K9me3 and repress many subfamilies of ERVs, it was not clear why IAPs were the main focus. For instance, families such as MusD/ETn, Gln and MMERVK10C all showed similar trends as IAP (Extended figure 3). It would be informative to at least consider those that have a substantial enrichment of H3.3.

IAP ERVs as a family are in quite exceptional in terms of KAP1 and Smardad1 recruitment, thus they provide the best example for interstitial heterochromatin.

It is important to note however, that the H3.3+H3K9me3 peaks shown in Fig 1d, 2a,b,c,d , 3a, 5a, b provide a representative sampling of elements from different ERV families. We have previously summarized this in Elsässer et. al. 2015 as Extended Data Figure 2

Distribution of H3.3 and H3K9me3 peaks amongst interspersed repeats

and have here performed a more detailed analysis:

Num repeat elements overlapping H3.3 + H3K9me3 peaks

Numer reflects the total number of overlaps, “frac” represents the fraction of elements for each repeat family that contain a H3.3+H3K9me3 peak.

It is true that ETn/MusD and ERVK10C elements may follow the same regulation, However, it is problematic to study them as a family if only ~25% contain H3.3+H3K9me3 peaks. See also Extended Data Figure 4a, while most elements are decorated with H3K9me3, a small number,

~5%, of annotated elements are highly transcribed (see Extended Data Figure 4a). Thus when analyzing repetitive sequences it becomes ambiguous if H3.3 is deposited within heterochromatin or within active elements.

3. With the analysis of ETn elements (Extended data figure 4), I have some concerns regarding the derived conclusions. ETn elements are far smaller and exist in fewer copies as IAPs, the qualitative comparison in panels A and B are not equivalent.

To clarify this, we have adjusted the height of the heatmaps to proportion.

It is unclear what the values of the heatmap are. It would be more informative to conduct a quantitative comparison on an individual elements level.

All tracks are scaled to a 1x genome coverage, so the blue graphs and heatmaps can be compared quantitatively. For additional clarity we have performed an enrichment analysis on the individual element +/- 1kb in the new panels Extended Figure 4b, d

Moreover, it was not explained clearly as to how the calculations of Extended data figure 4C was done. If copy number was not considered, then this would potentially create a bias. I don't doubt the result but the calculation is hard to understand. Also, it is worth noting that it has been long reported that some ETn/MusD elements have activity in ESCs and given that ETns are the non-autonomous versions of MusD, they are often considered together.

We did not mean to report this as a novel finding but thought it would be important to point out the heterogeneity within ETn as a rational not to overinterpret H3.3 enrichment at these elements since it could in part due to high transcriptional activity.

We have changed the representation of the transcript-level data now Extended Data Figure 4e, plotting individual elements rather than the mean expression of the family:

4. In figures 3E and Extended Figure 9, it looks as if the reduced mappability of the IAP internal sequences may be confounding the results. Given that uniquely mapped versus all mapped reads would give different results, it is difficult to conclude whether those that are losing H3.3 are actually gaining chromatin accessibility. Quantitative analysis on an individual element level would help. Also, consider showing the mappability of the IAP consensus sequence.

Figure 3e is including multimap reads, thus to be interpreted as an average over all elements. Figure 3f shows the same elements but considering only uniquely mappable read pairs. It is clear from the heatmap that the vast majority of instances experiences an increase in accessibility since an increase in signal is apparent also from unique reads at the edges/flanking regions of the element. At the same time, analysis of H3.3 by uniquely mappable reads in Extended Data Figure 9 shows that H3.3 is present in the flanking regions of almost all instances.

A separate heatmap for mappability is also added to Extended Data Figure 9:

Uniquely mapped reads (MQ > 10)

We have taken up the suggestion by the reviewer and added a per-element quantitation using uniquely mappable reads in Extended Data Figure 7e. It is important to note that the datapoints at the low end appear to have neither high accessibility nor H3.3, but this may simply reflect the low number of uniquely mappable reads. Importantly, almost all datapoints lie above the diagonal, thus most individual elements experience an increase in the absence of H3.3.

5. In addition to the issue of poor mappability, as the cells surveyed are asynchronous, I wonder if the change in chromatin accessibility is a feature of a heterogeneous population.

Heterogeneity can be expected on several levels; individual instances of the same repeat family may behave differently within the same cell or across cells in the population. However, since the average effect size we are observing in Figure 3e, 4c is large, it is unlikely that the change we are observing derives from a small subpopulation of cells or subset of instances.

6. Generally, there is a lack of quantitative comparisons. Statistical tests can be applied to data presented in Figure 1A, 1E, 2C, etc.

We have performed statistical tests and effect size estimations where appropriate to address this issue.

Minor Comments

1. Consider including H3.3 ChIP-seq track for Figure 1B.

Done

2. It's unclear what the 2 black and white columns for ESC/NSC H3.3-SNAP-0h show. The legends suggest they are peaks but I don't understand what data they present.

The additional heatmaps in Figure 1d are reflect intersections of each peak instance (line in the heatmap) with each the respective annotations (ChromHMM, IAPEz-int, ESC/NSC H3.3 SNAP 0h peaks from Fig 1c). E.g. a black line indicates coincidence of the peak with an IAPEz-int whereas a white line indicates the absence of such overlap.

3. Extended figure 5b was not mentioned in text. I believe it corresponds to Extended Figure 6 on page 7.

fixed

4. Several figures are missing proper labelling of axis, legends, etc. (Figure 2E and 3F, Extended figures 1C, 4A-B, 5A and 8)

fixed

5. More details need to be provided in the methods and analysis. I was unable to find experimental details as well as how some calculations were done. For instance, in Fig 1E, log₂ over input is unclear to me. However, it's also not explained in the methods.

We have added methods for then new experiments included in Figure 4 and 5 and have detailed the methods for reanalyzing public datasets in Figures 1-3. Regarding log₂ over input, we have normalized our bin-based data using the matching input dataset where available and then transformed the CHIP/input ratio by log₂

6. It's unclear why RPGC was used instead of typical RPKM.

RPKM is a useful measure for Transcript-level data but it lacks a meaningful unit for genomics data. Reads Per Genomic Content (RPGC) normalizes the data to a global average of 1 (thus sometimes called 1x Genome Coverage normalization). Thus, the Y axis of RPGC-normalized data can be easily interpreted: Values below indicate a depletion as compared to the genomic average, whereas values larger than 1 indicate an enrichment. E.g. an RPGC of 10 is a 10fold enrichment over the average genomic signal. We believe such information is crucial to judge the magnitude of enrichment in genomic plots and RPKM does not provide the same clarity.

7. The format of the manuscript appears to be different to others. The authors should break the results into sub-sections for easier reading.

done

Bibliography

Banaszynski, L.A., Wen, D., Dewell, S., et al. 2013. Hira-dependent histone H3.3 deposition facilitates PRC2 recruitment at developmental loci in ES cells. *Cell* 155(1), pp. 107–120.

Chronis, C., Fiziev, P., Papp, B., et al. 2017. Cooperative binding of transcription factors orchestrates reprogramming. *Cell* 168(3), pp. 442–459.e20.

Elsässer, S.J., Noh, K.-M., Diaz, N., Allis, C.D. and Banaszynski, L.A. 2015. Histone H3.3 is required for endogenous retroviral element silencing in embryonic stem cells. *Nature* 522(7555), pp. 240–244.

Fang, H.-T., El Farran, C.A., Xing, Q.R., et al. 2018. Global H3.3 dynamic deposition defines its bimodal role in cell fate transition. *Nature Communications* 9(1), p. 1537.

Gehre, M., Bunina, D., Sidoli, S., et al. 2020. Lysine 4 of histone H3.3 is required for embryonic stem cell differentiation, histone enrichment at regulatory regions and transcription accuracy. *Nature Genetics* 52(3), pp. 273–282.

Jang, C.-W., Shibata, Y., Starmer, J., Yee, D. and Magnuson, T. 2015. Histone H3.3 maintains genome integrity during mammalian development. *Genes & Development* 29(13), pp. 1377–1392.

Martire, S., Gogate, A.A., Whitmill, A., et al. 2019. Phosphorylation of histone H3.3 at serine 31 promotes p300 activity and enhancer acetylation. *Nature Genetics* 51(6), pp. 941–946.

Matsui, T., Leung, D., Miyashita, H., et al. 2010. Proviral silencing in embryonic stem cells requires the histone methyltransferase ESET. *Nature* 464(7290), pp. 927–931.

Rowe, H.M., Jakobsson, J., Mesnard, D., et al. 2010. KAP1 controls endogenous retroviruses in embryonic stem cells. *Nature* 463(7278), pp. 237–240.

REVIEWER COMMENTS

Reviewer #1 (Remarks to the Author):

Navarro et al. have addressed most of the comments raised, nevertheless there are a few concerns that need to be addressed.

1) The authors have added new data on smarcad1 knockout (Fig 4 and 5) and arrived at a conclusion that smarcad1 is a putative chromatin remodeler that evicts H3.3 nucleosomes. The data is interesting but it is not clear how smarcad1 was selected as a putative factor that drives H3.3 eviction across ERV regions. This could be better explained. The authors concluded that Smarcad1 evicts H3.3 nucleosomes, but there is a lack of direct evidence to support this claim. Moreover, smarcad1 has been found to be essential for ERV silencing (Sachs et al 2019 Nat Comm). A few concerns that need to be addressed. What happens to H3.3 nucleosomes and DNA accessibility at ERVs when only smarcad1 is knocked down? Do the authors expect enriched presence of H3.3 nucleosomes and heterochromatin mark H3K9me3 at ERVs in the absence of smarcad1? This needs to be investigated and compared with levels of H3.3 nucleosomes, histone modifications, transcription activity and DNA accessibility in H3.3 KO and H3.3 KO/Smarcad1 knockdown cells.

2) The authors showed that knockdown of Smarcad1 reverted DNA accessibility at IAP in H3.3 KO cells. How would the authors reconcile this finding with the finding from Sachs et al 2019 Nat Comm which smarcad1 is required for the silencing of ERV and lack of SMARCAD1 compromises heterochromatin at ERVs. How did Smarcad1 KO revert heterochromatin accessibility if H3.3 is absent at ERVs? Are H3.3 nucleosomes replaced with H3.1/2 nucleosomes in the absence of smarcad1? It is unclear how heterochromatin accessibility is reverted.

Reviewer #2 (Remarks to the Author):

The authors have addressed my concerns and I have no further comments.

Summary of updates:

- added new Extended Data Figure 12 with new ATAC-Seq data
- added new Figure 6, Extended Data Figures 13, 14 with new ChIP-Seq data
- amended result sections and edited the discussion section
- text changes in R2 are highlighted in red
- Updated GEO entry with new data
- we suggest to change the title (red highlight)

Reviewer #1 (Remarks to the Author):

Navarro et .al. have addressed most of the comments raised, nevertheless there are a few concerns that need to be addressed.

1) The authors have added new data on smarcad1 knockout (Fig 4 and 5) and arrived at a conclusion that smarcad1 is a putative chromatin remodeler that evicts H3.3 nucleosomes. The data is interesting but it is not clear how smarcad1 was selected as a putative factor that drives H3.3 eviction across ERV regions. This could be better explained.

We had discussed the rational in following section:

“It caught our attention that the chromatin remodeler Smarcd1 has recently been linked to IAP ERVs in mouse ESC 47. DAXX and Smarcd1 have been independently shown to interact with KAP1 but no biochemical or functional interaction between the two proteins has been reported 21,39,48. However, Smarcd1 has further been shown to have nucleosome sliding and eviction activity 49,50, thus representing an interesting candidate for mediating nucleosome turnover at ERVs. Reanalyzing Smarcd1 ChIP-Seq 47,51 showed that Smarcd1 specifically localized to H3.3+H3K9me3 peaks (Figure 5a)”

The authors concluded that Smarcd1 evicts H3.3 nucleosomes, but there is a lack of direct evidence to support this claim. Moreover, smarcd1 has been found to be essential for ERV silencing (Sachs et al 2019 Nat Comm). A few concerns that need to be addressed. What happen to H3.3 nucleosomes and DNA accessibility at ERVs when only smarcd1 is knocked down?

We had thought along the same lines and thus were already in the process of addressing both these questions with additional ATAC-Seq and CHIP-Seq experiments.

We have now shown in a second batch of ATAC-Seq experiments that Smarcd1 knockdown in wildtype ESC further reduces the already low DNA accessibility. While the effect size is minor given that the regions are heterochromatic to start with, this observation is fully in line with our mechanistic model (i.e. that, in wildtype ESC, H3.3 quickly replenishes evicted nucleosomes). We have included ATRX knockdown as additional control

Do the authors expect enriched presence of H3.3 nucleosomes and heterochromatin mark H3K9me3 at ERVs in the absence of smarcad1?

This needs to be investigated and compared with levels of H3.3 nucleosomes, histone modifications, transcription activity and DNA accessibility in H3.3 KO and H3.3 KO/Smarcad1 knockdown cells.

We have also performed H3.3 and H3K9me3 ChIP-Seq comparing wildtype and Smarcad1 KD cells, and H3.3 is significantly reduced by 40% compared to wildtype levels at H3.3+H3K9me3 peaks in Smarcad1 KD cells. Over IAP ERVs, H3.3 is significantly reduced by ~35%.

Figure 6

Again, this is in line with and further pinpoints our mechanistic model in which Smarcad1 drives nucleosome eviction, which triggers H3.3 deposition. Without Smarcad1 activity, there is no opportunity to incorporate H3.3 in the first place. The moderate reduction of H3.3 stems from two reasons: First, Smarcad1 knockdown does not completely abrogate its protein level. Second, even if Smarcad1 activity was completely abolished, the cessation of nucleosome turnover would lead to a retention of the existing histones, i.e. H3.3 would remain where it is at IAP ERVs and diluted only by replication. Thus, assuming maximally two cell divisions after knockdown takes effect, it would be expected that 25% or more H3.3 still remained.

2) The authors showed that knockdown of Smarcad1 reverted DNA accessibility at IAP in H3.3 KO cells. How would the authors reconcile this finding with the finding from Sachs et al 2019 Nat Comm which smarcad1 is required for the silencing of ERV and lack of SMARCAD1 compromises heterochromatin at ERVs.

We had discussed this in the following section

“Loss of H3.3, as well as loss of Smarcad1 have been shown to reduce H3K9me3 levels and KAP1 occupancy at IAP ERVs^{21,47} which suggests that the dynamic process reinforces rather than disrupts heterochromatin. Transient opening of chromatin could allow KRAB-ZFPs proteins to access the underlying DNA sequence and consequently amplify the repressive domain by recruitment of additional KRAB-ZFP/KAP1 corepressor complexes along the repetitive DNA sequence. Reassembly of nucleosomes with the replication-independent substrate histone H3.3

would subsequently be necessary to maintain a compact chromatinized state with high levels of H3K9me3. “

Of note, we did not observe a loss of H3K9me3 upon the 48h knockdown of Smarcd1. This is in contrast to previously established knowledge that stable knockdown of Smarcd1 reduces H3K9me3 (Sachs et. al. 2019) and H3.3 knockout reduces H3K9me3 at IAP ERVs (Elsässer et. al., 2015), and that loss of Smarcd1 or H3.3 leads to (moderate in comparison to SETDB1 or KAP1 knockout!) upregulation of ERVs (Sachs 2019, Elsässer, 2015, Hoelper et. al., 2017). Again, in the absence of Smarcd1-induced nucleosome eviction and turnover, H3K9me3 would be expected to be a very stable modification (little evidence for active demethylation). The 48h knockdown leaves limited time for the heterochromatin state to ‘erode’.

To speculate further on the mechanism of how loss of Smarcd1 leads to desilencing eventually, we would propose following hypothesis: KRAB-ZFP must transiently dissociate from the ERVs upon DNA replication and may compete with nucleosomes for their DNA binding sequence in the wake of replication. The Smarcd1 nucleosome remodeling activity may thus open chromatin for these factors to re-engage with their ERV targets and subsequently recruit KAP1 and associated heterochromatin factors. At the same time, H3.3 chromatin assembly is then required to replenish nucleosomes, thus rationalizing why both Smarcd1 and H3.3 are required to maintain high H3K9me3 levels.

While we focus here on the discovery of a dynamic heterochromatin state and identify the major factors involved, we acknowledge that further work will be necessary to follow up on why this dynamic heterochromatin state exists in the first place and if or not it is generally repressing transcriptional output, or in other instances would also promote it. However we strongly believe these questions are beyond the scope of the current manuscript.

How did Smarcd1 KO revert heterochromatin accessibility if H3.3 is absent at ERVs? Are H3.3 nucleosomes replaced with H3.1/2 nucleosomes in the absence of smarcd1? It is unclear how heterochromatin accessibility is reverted.

As discussed above, our mechanistic model (Figure 6) explains this, and all previous and new experiment support his model: Smarcd1 remodeling activity creates accessible DNA by evicting nucleosomes, but this is not detected in wildtype cells because nucleosomes are immediately replenished with H3.3. Only in the absence of H3.3, we observe the consequences of Smarcd1 activity, open DNA.

Smarcd1 knockdown attenuates this nucleosome eviction activity. Without Smarcd1 remodeling activity, no histones are evicted and no accessible DNA is created in the first place. Consequently no free DNA is available to incorporate H3.3. Neither is there an active removal of H3.3 or back-exchange to H3.1/2, but it is expected that CAF1 introduces newly synthesized H3.1/2 to assemble chromatin in the wake of the replication fork and thus H3.3 is passively diluted out over cell divisions if Smarcd1 activity ceases.

REVIEWERS' COMMENTS:

Reviewer #1 (Remarks to the Author):

The authors have addressed all the comments and concerns. Data from the additional ATAC-Seq and ChIP-Seq experiments (from SMARCAD1 knockdown) have helped to clarify the relationship between H3.3, SMARCAD1 in maintaining heterochromatin and Chromatin accessibility at repetitive elements.